# Efficient Fully Single-Loop Variance Reduced Methods for Stochastic Bilevel Optimization

## Abstract

Stochastic Bilevel Optimization (StocBO) has gained traction given its unique nested structure, which is increasingly popular in machine learning areas like meta-learning and hyperparameter optimization. A recent innovation by Dagréou et al. (2022) provided a unified single-loop framework for finite-sum StocBO. This presented the SABA method, a SAGA-type approach, achieving an iteration complexity of $\mathcal{O}({(m+n)^{3/2}}/{T})$ and a memory cost of $\mathcal{O}((m+n)(d+p))$. In this context, $m$ and $n$ symbolize the finite sum counts for the outer and inner-level tasks, while $d$ and $p$ describe their parameter dimensions. However, a drawback surfaces with memory consumption, especially with significantly large values of $m$ or $n$. In response to this, we present the SBO-LSVRG, an adept solution inspired by Loopless-SVRG (LSVRG) (Kovalev et al., 2020). This avant-garde method not only achieves the desired iteration complexity but also substantially trims the memory cost to a leaner $\mathcal{O}(d+p)$. To our awareness, this paper pioneers in illustrating, from a theoretical lens, the application of LSVRG to bilevel optimization, particularly in non-convex realms. Furthermore, our variance-reduced method, SBO-LSVRG, excels with an optimal convergence speed. Comprehensive experiments validate the efficiency of our proposed approach.

## 1 Introduction

Bilevel optimization is increasingly attracting attention in machine learning due to its broad applications, such as in meta-learning (Franceschi et al., 2018; Ji et al., 2020; Rajeswaran et al., 2019) and hyperparameter tuning (Bengio, 2000; Bertrand et al., 2020; Pedregosa, 2016). In this paper, we primarily focus on the standard Stochastic Bilevel Optimization (StoBO) problem, expressed in the finite-sum form:

$$\min_{x\in\mathbb{R}^d}\Phi(x)=\frac{1}{m}\sum_{j=1}^{m}F_j(y^\star(x),x),\quad \text{s.t.}\quad y^\star(x)\in\arg\min_{y\in\mathbb{R}^p}\frac{1}{n}\sum_{i=1}^{n}G_i(y,x),\qquad \text{(StoBO)}$$

where functions $F_j$ and $G_i$ represent outer- and inner-level problems, indexed by $j \in [m]$ and $i \in [n]$ respectively. The intuitive approach involves running multiple local steps to obtain the optimal $y^\star(x)$ or a good approximation thereof. We then use this to update the outer loop (Ghadimi & Wang, 2018; Ji et al., 2021; Arbel & Mairal, 2021; Yang et al., 2021a). However, this kind of two-loop method often falls short in terms of efficiency during implementation. For this reason, simultaneously updating the outer- and inner-level parameters typically yields better performance, as seen in SVRB Guo et al. (2021a), SBFW Akhtar et al. (2021), FSLA Li et al. (2022), and SABA Dagréou et al. (2022).

By leveraging the implicit function theorem under appropriate assumptions, we can utilize the general single-loop framework suggested by Dagréou et al. (2022). We define $v$ to approximate the inverse Hessian, enabling us to formulate any algorithm based on the following update rules: for each iteration $t$, in the context of a single node, we have:

Table 1: Summary of variance reduced methods for solving StoBO.

| Method | Loops | VR Method[a] | Iteration Complexity | Space # Round |
|---|---|---|---|---|
| SVRB Guo et al. (2021a) | Single | STORM | $\tilde{\mathcal{O}}(\epsilon^{-3})$ | $\mathcal{O}(d + p)$ |
| VRBO Yang et al. (2021b) | Double | SARAH | $\tilde{\mathcal{O}}(\epsilon^{-1.5})$ | $\mathcal{O}(d + p)$ |
| SABA Dagréou et al. (2022) | Single | SAGA | $\mathcal{O}\left((m + n)^{2/3}\epsilon^{-1}\right)$ | $\mathcal{O}((m + n)(d + p))$ |
| SBO-LSVRG (**Ours**) | Single | L-SVRG | $\mathcal{O}\left((m + n)^{2/3}\epsilon^{-1}\right)$ | $\mathcal{O}(d + p)$ |

[a] The term "Variance Reduction (VR) Method" refers to techniques employed to reduce the variance resulting from stochastic gradient estimation. The following table outlines the primary concept behind each paper. It is important to note that the methods may not be exactly the same, as the objective StoBO introduces novel elements.

$$
\begin{aligned}
y^{t+1} &= y^t - \rho^t D_y^t, \quad s.t. \quad D_y^t = \nabla_y G(y, x) \\
v^{t+1} &= v^t - \rho^t D_v^t, \quad s.t. \quad D_v^t = \nabla_{yy}^2 G(y, x)v + \nabla_y F(y, x) \\
x^{t+1} &= x^t - \gamma^t D_x^t, \quad s.t. \quad D_x^t = \nabla_{xy}^2 G(y, x)v + \nabla_x F(y, x),
\end{aligned}
\tag{1}
$$

where $D_y^t, D_v^t, D_x^t$ are the updating directions for the inner model, the Hessian inverse approximation parameter, and the outer model at iteration $t$. $\nabla_{yy}$ and $\nabla_{xy}$ denotes the Hessian and Jacobian matrix with respect to parameters $x$ and $y$. Given that the inner problem with respect to $y$ and the linear system with respect to $v$ exhibit similar conditioning, we follow the approach similar to SABA from Dagréou et al. (2022), wherein we employ the same stepsize for updating both $y$ and $v$.

SABA, as proposed by Dagréou et al. (2022), introduced the following SAGA-like updates, as noted in Defazio et al. (2014). Let's say the update rules in eq. (1). Without lose of generality, let's consider the update for $y$. Assume we select $i$ from $[n]$, the brief idea of Defazio et al. (2014) in SABA is we consider the following update:

$$
g_y^t = \nabla_y G_i(y^t, x^t) - \nabla_y G_i(w_{y,i}^t, w_{x,i}^t) + \frac{1}{n}\sum_{j=1}^{n} \nabla_y G_j(w_{y,j}^t, w_{x,j}^t),
\tag{2}
$$

where $w_{y,j}^t, w_{x,j}^t$ represent the stored local vectors for node $i$. Suppose $z \in \{y, x\}$, $w_{z,j}^{t+1}$ is updated as follows:

$$
w_{z,j}^{t+1} = \begin{cases} z^t & j = i \\ w_{z,j}^t & j \neq i. \end{cases}
\tag{3}
$$

For obtaining $g_y^t$ in Equation (2), the corresponding additional memory space cost is $n \times p$. Considering the whole update for all the $y$, $v$ and $x$, the total memory cost is $(m + n)d + (m + 2n)p = \mathcal{O}((m+n)(d+p))$. In practical scenarios where $m$ or $n$ is large, the total space complexity becomes substantially high.

Our paper primarily focuses on reducing the space complexity of single-loop methods while still maintaining a satisfactory level of performance. Additionally, we are keen to extend the single-loop framework suggested by Dagréou et al. (2022) in light of potential connections to federated learning and minimax optimization.

**Contributions.** Our contributions can be summarized as follows:

- a) Taking inspiration from the loopless stochastic variance reduced gradient estimator (L-SVRG) proposed by Kovalev et al. (2020), we propose an innovative method, SBO-LSVRG, which achieves state-of-the-art iteration complexity. When compared with SABA, our approach has a significantly lower memory cost that remains constant in relation to $m$ and $n$, making it more desirable. This approach is far from trivial.

- b) To our knowledge, SBO-LSVRG is the first paper to consider L-SVRG for problems with a nested structure. Furthermore, it is the only approach that addresses non-convex L-SVRG (where the outer-problem exhibits the L-SVRG style, but does not necessarily have to be convex). We compared related methods in Tab.1

- c) We establish the link between our method and related areas, such as federated learning and minimax optimization, and we provide a theoretical analysis for both of these areas.

- d) We conduct thorough experiments to validate the efficiency of our proposed methods.

## 2 PRELIMINARY

In this section, we state the main assumptions and recall some useful lemmas.

### 2.1 ASSUMPTIONS

We first state the standard assumptions on the functions $G$ and $F$.

**Assumption 1 (Convexity)** *The inner-level function $G(\cdot, x)$ is $\mu_G$-strongly convex for any $x \in \mathbb{R}^d$.*

The strong convexity of the inner problem guarantees the existence and uniqueness of solutions for optimizing $G$ for any $x \in \mathbb{R}^d$. Such an assurance is a staple when analyzing bilevel optimization. Importantly, note that the outer-level function $F$ is not mandated to be strongly convex. This suggests that our configuration is of a strongly-convex-nonconvex nature. We now delve into the standard Lipschitz-smoothness.

**Assumption 2 (Smoothness)** *Here we consider the smoothness for both the inner- and outer-level functions $G$ and $F$:*

- *$G$ is three times continuously differentiable on $\mathbb{R}^p \times \mathbb{R}^d$. The derivatives $\nabla G, \nabla^2 G$ and $\nabla^3 G$ are Lipschitz continuous in $(y, x)$ with Lipschitz constants $L_1^G, L_2^G$ and $L_3^G$.*

- *$F$ is twice continuously differentiable. The derivatives $\nabla F$ and $\nabla^2 F$ are Lipschitz continuous in $(y, x)$ with constants $L_1^F$ and $L_2^F$.*

- *For all $i \in [n]$ and $j \in [m]$, the gradients $\nabla G_i, \nabla F_j, \nabla_{yy}^2 G_i$ and $\nabla_{xy}^2 G_i$ are Lipschitz continuous in $(y, x)$.*

Furthermore, to ensure the boundedness of $v^\star$, we follow a similar approach to Dagréou et al. 2022 Assumption 3.3-3.7, where we bound the gradient of the outer problem $F$ with respect to $y$. Additionally, to maintain equivalent complexity for the single-level problem, it's necessary to bound the expected norm of the directional gradient $D_x^t$. These two bounding conditions are articulated in the subsequent assumption.

**Assumption 3 (Variance Bounds)** *There exists $C_F > 0$ and $B_x > 0$ suc that for any $x$ and $t$, we have $\|\nabla_y F(y^\star(x), x)\| \le C_F$ and $\mathbb{E}_t\left[\|D_x^t\|^2\right] \le B_x^2$.*

### 2.2 USEFUL LEMMAS

Next, we present two useful lemmas related to bound the directional gradients $D_y, D_v$ and $D_x$, and the smoothness constant of function $\Phi$.

**Lemma 1 (Dagréou et al. 2022, Lemma 3.4)** *Let Assumption 1 and 2 hold, there exist constants $L_y, L_v$ and $L_x$ such that*

$$\|D_y(y, v, x)\|^2 \le L_y^2 \|y - y^\star(x)\|^2,$$
$$\|D_v(y, v, x)\|^2 \le L_v^2(\|y - y^\star(x)\|^2 + \|v - v^\star(x)\|^2),$$
$$\|D_x(y, v, x)\|^2 \le L_x^2(\|y - y^\star(x)\|^2 + \|v - v^\star(x)\|^2),$$

where $L_y = L_1^G, L_v = L_x = \sqrt{2} \max \left( \frac{L_2^G C_F}{\mu_G} + L_1^F, L_1^G \right)$.

**Lemma 2 (Ghadimi & Wang 2018, Lemma 2.2)** *Let Assumption 1 and 2 hold, the function $\Phi$ is $L^\Phi$-smooth for some $L^h > 0$, where*

$$L^\Phi = L_1^F + \frac{2L_1^F L_2^G + C_F^2 L_2^G}{\mu_G} + \frac{L_{11}^G L_1^G C_F + L_1^G L_2^G C_F + \left(L_1^G\right)^2 L_1^F}{\mu_G^2} + \frac{\left(L_1^G\right)^2 L_2^G C_F}{\mu_G^3}.$$

## 3 CONVERGENCE ANALYSIS OF SBO-LSVRG

In this section we propose SBO-LSVRG and present its convergence analysis.

### 3.1 SBO-LSVRG

Next we present the details of our proposed method SBO-LSVRG. The general idea is that we consider the L-SVRG-like (Kovalev et al., 2020) updates for equation StoBO. We already present the directional gradient for $y, v, x$ in Equation (1), the main idea is that we introduce a global variable for each full gradient computation to avoid high computation costs. Without loss of generality, we take $D_y^t = \nabla_y G(y, x)$ as an example. Suppose at iteration $t$, we sample $i \in [n]$ and $j \in [m]$. For simplexity, we are interested in the uniform sampling which means $q_y = 1/n, q_x = 1/m$. Then the updating rule in SBO-LSVRG is

$$D_y^t = \nabla_y G_i(y^t, x^t) - \nabla_y G_i(w_y^t, w_x^t) + \nabla_y G(w_y^t, w_x^t). \tag{4}$$

Similar to Defazio et al. (2014) and Dagréou et al. (2022), we can view out method as having two "parallel" memories for each variable set $(w_y^t, w_v^t, w_x^t)$ for $i \in [n]$ corresponding to calls in $G$ and $(w_y'^t, w_v'^t, w_x'^t)$ for $j \in [m]$ corresponding to calls to $F$, which reflects on the sampling of $G_i$ and $F_j$, respectively. We can formulate the update as follows

$$\{w_y^{t+1}, w_v^{t+1}, w_x^{t+1}\} = \begin{cases} \{w_y^t, w_v^t, w_x^t\} & \text{with probability}(1 - q_y)(1 - q_x) \\ \text{others} & \text{otherwise.} \end{cases} \tag{5}$$

Then, the updating for the other two directional gradients for $v$ and $x$ are as follows

$$\begin{aligned} D_v^t &= \nabla_{yy}^2 G_i(y^t, x^t)v^t - \nabla_{yy}^2 G_i(w_y^t, w_x^t)w_v^t + \nabla_{yy}^2 G(w_y^t, w_x^t)w_v^t \\ &\quad + \nabla_y F_j(y^t, x^t) - \nabla_y F_j(w_y'^t, w_x'^t) + \nabla_y F(w_y'^t, w_x'^t) \\ D_x^t &= \nabla_{xy}^2 G_i(y^t, x^t)v^t - \nabla_{xy}^2 G_i(w_y^t, w_x^t)w_v^t + \nabla_{xy}^2 G(w_y^t, w_x^t)w_v^t \\ &\quad + \nabla_x F_j(y^t, x^t) - \nabla_x F_j(w_y'^t, w_x'^t) + \nabla_x F(w_y'^t, w_x'^t) \end{aligned} \tag{6}$$

Combining the updating rules in Equation (5) and Equation (6), now its clear that we will have probability $(1 - q_y)(1 - q_x)$ with constant gradient computes and the rest of probability with the same number of gradient computes as SABA and SOBA. In practice, $m$ and $n$ are supposed to be very large, considering the uniform probability $q_y = 1/n$ and $q_x = 1/m$, and then $(1 - q_y)(1 - q_x)$ will be very close to 1, which means we always have less gradient computes compared with SABA and SOBA Besides, our space memory cost is constant as we only need to store six global variables $(w_y^t, w_v^t, w_x^t, w_y'^t, w_v'^t, w_x'^t)$.

It's obvious to know they are all unbiased estimators w.r.t. $i$ and $j$, e.g.,

$$\begin{aligned} \mathbb{E}_t \left[ D_y^t(y^t, v^t, x^t) \right] &= \sum_{i=1}^n \frac{1}{n} \nabla_y G_i(y^t, x^t) - \sum_{i=1}^n \frac{1}{n} \nabla_y G_i(w_y^t, w_x^t) + \nabla_y G(w_y^t, w_x^t) \\ &= \frac{1}{n} \sum_{i=1}^n \nabla_y G_i(y^t, x^t) := D_y(y^t, v^t, x^t). \end{aligned} \tag{7}$$

We present the algorithm details of SBO-LSVRG in Algorithm 1.

---

**Algorithm 1** SBO-LSVRG: Loopless SVRG for optimizing Equation (StoBO)

---

1: **Parameters:** Initialize $y^0 \in \mathbb{R}^p, z^0 \in \mathbb{R}^p, x^0 \in \mathbb{R}^d$, number of iterations $T$, step size sequences $\{\alpha_y^t\}_{t<T}, \{\alpha_v^t\}_{t<T}$ and $\{\alpha_x^t\}_{t<T}$.
2: **for** $t = 0, 1, \ldots, T-1$ **do**
3:      Compute the directional gradient $D_y^t$ for $y$ by Equation (4).
4:      Compute the directional gradient $D_x^t$ and $D_v^t$ by Equation (6).
5:      Update $y : y^{t+1} = y^t - \rho^t D_y^t$.
6:      Update $v : v^{t+1} = v^t - \rho^t D_v^t$.
7:      Update $x : x^{t+1} = x^t - \gamma^t D_x^t$.
8:      Update stored global memory by Equation (5).
9: **end for**

---

## 3.2 CONVERGENCE ANALYSIS

Now, we are ready to present our main theory.

**Theorem 1 (Convergence of SBO-LSVRG.)** *Assume Assumptions 1, 2, 3 hold. Let $x^t$ the iterates of SBO-LSVRG. Following the updating rules Equations (4) and (6) to optimize StoBO. Then,*

$$\frac{1}{T} \sum_{t=1}^{T} \mathbb{E}\left[\left\|\nabla h(x^t)\right\|^2\right] = \mathcal{O}\left((m+n)^{2/3}/T\right).$$

Theorem 1 emphasizes that the expected second moment of the conjugated gradient will be bounded by $\mathcal{O}((m+n)^{2/3}/T)$, where $m$ and $n$ is the number of the finite-sum for the outer and the inner problems. This result leads to the convergence rate $\mathcal{O}(\epsilon^{-1})$, which is optimal in stochastic bilevel optimization. Given that the oracle calls per iteration amount to $\mathcal{O}(1)$, it follows that the sample complexity is $\mathcal{O}(\epsilon^{-1})$. To the best of our knowledge, this sample complexity is also optimal. Next, our focus will shift to exploring the convergence rates of downstream tasks. In this context, single-level optimization with the finite-sum form is a standard approach in federated learning. Additionally, minimax optimization is a popular formulation for the training of generative adversarial networks.

**Corollary 1 (Single-level optimization)** *Although the work of Kovalev et al. (2020) reveals that the iteration complexity of L-SVRG, when dealing with a $\mu$-strongly convex problem and employing uniform sampling, is $\mathcal{O}\left(\max\left\{\frac{L_{\max}}{\mu}, n\right\} \log \epsilon^{-1}\right)$, the rate under nonconvex conditions remains unclear. We initially introduce a rate of $\mathcal{O}(n^{2/3}\epsilon^{-1})$ in our analysis by setting $\rho^t \equiv 0$.*

Our theory is general and can also be extended to minimax optimization, which is the special case by letting $G := \frac{1}{n}\sum_{i=1}^{n} G_i = -F := \frac{1}{m}\sum_{i=1}^{m} F_j$ and assuming $m = n$:

$$\min_{x \in \mathbb{R}^d} \Phi(x) := \frac{1}{n} \max_{y \in \mathbb{R}^p} \sum_{i=1}^{n} F_i(y, x) \tag{Minimax}$$

**Corollary 2 (Minimax optimization)** *The iteration complexity for solving the minimax problem Minimax under strongly-convex setting is $\mathcal{O}(n^{2/3}\epsilon^{-1})$.*

It's clear that bilevel optimization presents more challenges than minimax optimization, a fact also theoretically established in Ji & Liang (2023). Our proof remains consistent when addressing minimax optimization, leading to the rate of $\mathcal{O}(n^{2/3}\epsilon^{-1})$. This aligns with the known optimal factor of $-1$ in $\epsilon$ (Zhang et al., 2019). An intriguing avenue for future research could involve reformulating the existing proof for bilevel optimization to achieve a tighter bound for the minimax problem.

Next we consider the theoretical results based on a popular Polyak- Lojasiewicz (PL) Inequality, which is always useful to prove the linear convergence of gradient descent methods.

**Assumption 4 (Polyak- Lojasiewicz (PL) Inequality)** *Suppose $\Phi^{\inf}$ is the bound from below such that $\Phi(x) \geq \Phi^{\inf}$ for all $x \in \mathbb{R}^d$. Then there exists $\mu_\phi > 0$ such that $\forall x \in \mathbb{R}^d$, $\Phi(x) - \Phi^{\inf} \leq \frac{1}{2\mu_\phi}\|\nabla\Phi(x)\|^2$.*

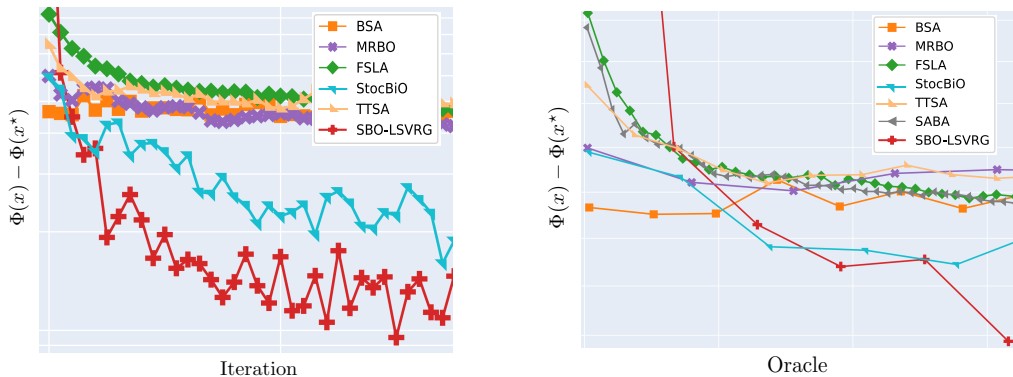

Figure 1: Testing accuracy on the covtype dataset.

**Theorem 2 (Convergence of SBO-LSVRG under PL)** *Assume Assumptions 1, 2, 3, 4 hold. Let $x^t$ the iterates of SBO-LSVRG and $c' = \min(\Phi_h, \frac{1}{16P'})$ with $P'$ and the learning rate $\gamma'$ specified in the appendix. Given $C^0$ depending on the initialization of $y, v, x$. Following the updating rules Equation (4) and Equation (6) to optimize StoBO. Then,*

$$\mathbb{E}\left[\Phi^T\right] - \Phi^{\inf} = (1 - c'\gamma')^T(\Phi^0 - \Phi^{\inf} + C^0).$$

Theorem 2 suggests the linear convergence of SBO-LSVRG. Drawing parallels with the analysis presented in Theorem 1, we can seamlessly derive the linear convergence of SBO-LSVRG for both minimax and single-layer finite-sum optimization tasks.

## 4 EXPERIMENTS

**Baselines.** We aimed to explore popular and easily implementable baselines that resonate with our objectives. In our experiments, we evaluated two-loop solvers: BSA (Ghadimi & Wang, 2018) and StocBiO (Ji et al., 2021), as well as one-loop solvers: TTSA (Hong et al., 2020), MRBO Yang et al. (2021a), FSLA (Li et al., 2022), and SABA (Dagréou et al., 2022).

### 4.1 HYPERPARAMETERS SELECTION

Our primary objective is to identify optimal hyperparameters for the $\ell^2$ logistic regression, emphasizing the determination of regularization parameters. We employ the Covtype dataset (Blackard, 1998) for this purpose. The inner function, denoted as $G$, is expressed as

$$G(x, z) = \frac{1}{n}\sum_{i=1}^{n}\ell(d_i; z) + \mathcal{R}(x, z),$$

where $d_1, \ldots, d_n$ represent training data samples, $z$ is the machine learning model's parameter, and the loss function $\ell$ quantifies the accuracy with which the parameter $z$ predicts data $d_i$.

The function also incorporates a regularization term, $\mathcal{R}$, parameterized by the regularization strengths $x$. This regularization encourages a specific structure on the parameters $z$. Conversely, the outer function, $F$, is the unregularized loss over unseen data and is defined as

$$F(x, z) = \frac{1}{m}\sum_{j=1}^{m}\ell(d'_j; z),$$

where $d'_1, \ldots, d'_m$ are samples distinct from the training set but from the same underlying dataset.

The Covtype dataset facilitates the logistic regression task. In this context, data samples are denoted as $d_i = (a_i, y_i)$, where $a_i \in \mathbb{R}^p$ are the features and $y_i \in \{-1, 1\}$ signifies the binary target. For this regression problem, the loss is given by

$$\ell(d_i, z) = \log(1 + \exp(-y_i a_i^T z)),$$

and the regularization is defined as

$$\mathcal{R}(x, z) = \frac{1}{2} \sum_{j=1}^{p} \exp(x_j) z_j^2,$$

where each coefficient in $z$ is independently regularized by the strength $\exp(x_j)$.

The dataset comprises $581,012$ samples, each characterized by $p = 54$ features, and spanning across 7 distinct classes. For our experiments, we utilized $n = 371,847$ samples for training, $m = 92,962$ samples for validation, and the remaining $116,203$ samples for testing. We applied a multiclass logistic regression to this dataset, designating one hyperparameter for each class.

We showcase the testing accuracy results in Figure 1. The left-hand side of Figure 1 displays a comparative analysis of the absolute distance between the current value and the optimal. The $x$-axis represents the number of iterations. Notably, when contrasted with prevalent baselines, our proposed SBO-LSVRG not only converges more swiftly but also settles closer to the optimal value.

Furthering our examination, the right-hand side of Figure 1 quantifies the temporal and spatial complexities associated with each baseline. More explicitly, let $\tau_a$ denote the time usage and $\tau_b$ symbolize the space consumption. We characterize the presented oracle as $\tau_a + \lambda \tau_b$, where $\lambda > 0$ serves as a weighting factor harmonizing the impacts of both time and space expenditures. In our experiments, we've set $\lambda = 0.001$.

A prominent observation is that our SBO-LSVRG consistently outperforms other baselines. Our keen interest gravitates towards its juxtaposition with SABA, given both share an identical optimal convergence rate. Yet, SABA exhibits a pronounced lag, chiefly attributable to its substantial space cost—a finding congruent with our theoretical predictions.

## 4.2 DATA HYPER-CLEANING

Our next endeavor involves applying the concept of data hyper-cleaning, as introduced by Franceschi et al. (2017), to the `MNIST` dataset (LeCun, 1998). We segregate the dataset into three distinct sets: a training set $(d_i^{\text{tr}}, y_i^{\text{tr}})$ consisting of 20,000 samples, a validation set $(d_i^{\text{val}}, y_i^{\text{val}})$ with 5,000 samples, and a test set holding 10,000 samples. The labels, denoted as $y$, span values from 0 to 9, and the sample features, represented by $x$, have a dimensionality of 784.

Within the training set, each sample faces potential "corruption" with a probability $p$. A sample is deemed corrupted if its original label $y_i$ gets substituted by a random label from the range $0 - 9$. Notably, the validation and test samples remain pristine, free from any corruption.

The crux of data hyper-cleaning is to meticulously train a multinomial logistic regression on the training set, with the overarching goal to discern a distinct weight for each training sample. Ideally, the weights attributed to corrupted samples should gravitate towards 0.

This principle finds its expression in the bilevel optimization problem delineated by equation StoBO. Herein, $F(\theta, \lambda) = \frac{1}{m} \sum_{i=1}^{m} \ell(\theta d_i^{\text{val}}, y_i^{\text{val}})$ and $G(\theta, \lambda) = \frac{1}{n} \sum_{i=1}^{n} \sigma(\lambda_i) \ell(\theta d_i^{\text{tr}}, y_i^{\text{tr}}) + C_r |\theta|^2$, where $\ell$ stands for the cross-entropy loss, while $\sigma$ signifies the sigmoid function.

In this construct, $\theta$ operates as the inner variable, manifesting as a $10 \times 784$ matrix. Conversely, $\lambda$, the outer variable, is a vector spanning a dimension of $n_{\text{tr}} = 20,000$.

For our analysis, we adopted varying corruption rates, the outcomes of which are visualized in Figure 2. A distinctive trait emerges for our hyper-cleaning method: the performance on the test dataset exhibits heightened stability relative to other baselines. For instance, while alternatives like TTSA exhibit pronounced variance—especially at a corruption ratio of 0.5 on MNIST—our approach remains consistently stable. Although stocBiO delivers commendably in hyper-parameter selection tasks, its convergence is sporadic, especially under conditions like a corruption ratio of 0.9. In stark contrast, our method not only converges briskly but also ensures that the convergence point remains both stable and proximate to the optimal solution.

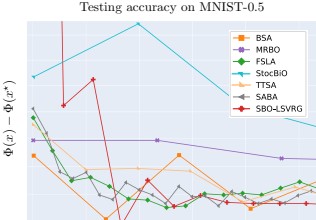 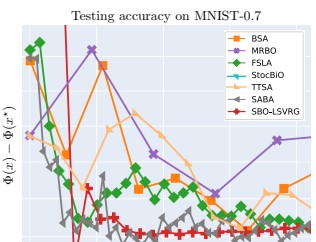 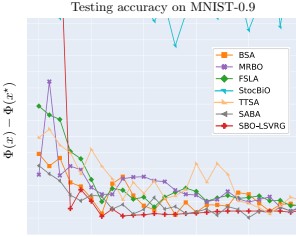

Figure 2: Testing accuracy on the MNIST dataset. The x-axis represents the combined consumption of time and space. Given the time usage $\tau_a$ and the memory space consumption $\tau_b$, with the balancing factor $\lambda$ set to 0.001, the x-axis is quantified as $\tau_a + \lambda\tau_b$.

## 5    RELATED WORKS

Bilevel optimization presents a higher level of complexity compared to single-level optimization due to its nested structure. Under specific assumptions, and leveraging the implicit function theorem from Griewank & Walther (2008), we can obtain the hypergradient for the outer problem as:

$$\nabla\Phi(x) = \nabla_x F(y^\star(x), x) - \nabla^2_{xy}G(y^\star(x), x)\left[\nabla^2_{yy}G(y^\star(x), x)\right]^{-1}\nabla_y F(x, y^\star).$$

A key challenge lies in computing this hypergradient directly, as it requires intricate computation of the involved Hessian, its inverse, and the Jacobian matrix. Stochastic bilevel optimization as outlined in eq. (StoBO) proves even more challenging, given the lack of a straightforward unbiased hypergradient estimator and the ongoing exploration of variance reduction methods for stochastic gradient estimators.

To calculate it efficiently, three prevalent general approaches are utilized: Back Propagation through iterations (BP), Neumann series (NS), and Conjugate gradient (CG). BP is formulated based on the chain rule, while NS and CG are constructed on varying strategies for approximating the inverse Hessian $\left[\nabla^2_{yy}G(y^\star(x), x)\right]^{-1}$. The study Li et al. (2022) introduced a general representation that encompasses all the mentioned approaches as special cases.

Given the nested structure in Stochastic Bilevel Optimization (StoBO), two distinct strategies naturally emerge. The first involves a two-loop approach, where the optimal or near-optimal $y^\star(x)$ is first obtained, after which $x$ is updated accordingly (Ghadimi & Wang, 2018; Ji et al., 2021; Yang et al., 2021b; Arbel & Mairal, 2021; Ji & Liang, 2023). A more direct single-loop strategy proposes updating all parameters simultaneously (Hong et al., 2020; Guo et al., 2021b; Yang et al., 2021a; Chen et al., 2022; Khanduri et al., 2021; Guo et al., 2021a; Akhtar et al., 2021; Li et al., 2022). As the two-loop methods tend to be inefficient due to the separation in the optimization design, we focus our attention on the single-loop optimization method. Our study is motivated by the novel single-loop network proposed in Dagréou et al. (2022).

In order to reduce the noise introduced by stochastic estimation in StoBO, variance reduction techniques are also utilized (Cutkosky & Orabona, 2019; Defazio et al., 2014; Kovalev et al., 2020). We provide a more comprehensive discussion of existing variance reduction methods in the Appendix. Our interest lies in the types and implementation of variance reduction methods within StoBO.

Due to the task's challenging nature, only a few works have addressed this issue (Guo et al., 2021a; Yang et al., 2021b; Dagréou et al., 2022). The SVRB algorithm contemplates a single-loop method, drawing inspiration from the technique of STORM (Cutkosky & Orabona, 2019), which employs a variant of the momentum term akin to Adam (Kingma & Ba, 2015). Concurrently, VRBO leverages SARAH (Nguyen et al., 2017) and achieves state-of-the-art iteration complexity among double-loop methods.

Rather than incorporating momentum-based variance reduction techniques to solve eq. (StoBO), the SABA algorithm Dagréou et al. (2022) attained state-of-the-art performance by using a simpler SAGA-like (Defazio et al., 2014) update. Nevertheless, SABA lacks memory efficiency due to the

need to store a large number of local weights. When the number of distribution parameters $m$ and $n$ for the outer- and inner-level problems are significant, SABA becomes highly cost-intensive.

Motivated by L-SVRG (Kovalev et al., 2020), we propose a new algorithm termed SBO-LSVRG, which substantially reduces the memory cost of SABA while achieving optimal iteration complexity.

A concurrent work, SRBA (Dagréou et al., 2023), also builds on the single-loop framework proposed by Dagréou et al. (2022) and utilizes the SARAH recursive structure for updating stochastic gradient estimates. Nonetheless, the recursive structure of SRBA leads to an increased sample complexity per iteration. Setting the number of inner iterations $p - 1 = m + n - 1$ yields a sample complexity per iteration of $\mathcal{O}(m + n)$, leading to a total sample complexity at a stationary point of $\mathcal{O}((m + n)^{1/2}\epsilon^{-1} \vee (m + n))$ as detailed in Corollary 3.6 in SRBA. In practical scenarios with significantly large $m$ or $n$, the sample complexity becomes $\mathcal{O}(m + n)$. In contrast, our method achieves a sample complexity of $\mathcal{O}((m+n)^{2/3}\epsilon^{-1})$. Especially when $m$ or $n$ is large, our method proves more efficient by a factor of $(m + n)^{1/6}$. Furthermore, SRBA's analysis right below Corollary 3.6 reveals that for large $m + n$, the sample complexity approaches $\mathcal{O}(\epsilon^{-2})$, which is suboptimal with respect to $\epsilon$. In conclusion, as $m + n$ becomes large (indicating an increase in data volume), the memory space required in SABA for storing additional status vectors becomes substantial and increases linearly. Meanwhile, SRBA's sample complexity deviates from the optimal $\mathcal{O}(\epsilon^{-1})$. Our method, in contrast, is more practical for large-scale data scenarios and effectively addresses these issues.

## 6 FUTURE WORK

One interesting future work is the experiments on Federated Learning (FL) tasks. FL is a burgeoning framework in machine learning that allows multiple clients to perform computations on their own private data, while periodically communicating with a remote server. The StoBO problem outlined in StoBO can be adapted to a federated learning scenario by setting $m = n$, where $n$ represents the number of clients.

One intriguing interpretation of bilevel optimization for federated learning could involve personalization, akin to what was presented in Hanzely et al. (2023). Here, we optimize the following objective:

$$\min_{\omega} F(\omega, \beta) := \frac{1}{m} \sum_{i=1}^{m} F_j(\omega, \beta_j),$$

In this scenario, $\beta_j$ is a specific optimal value for personalization. For instance, works such as Gasanov et al. (2021) and Kai et al. (2023) proposed that this could be a local optimum. This concept could be expanded to include the interpolation of the inner and outer output within a nested structure. Here, fine-tuning would occur post-acquisition of the global $\omega$, thereby updating the local $\beta$. Exploring the performance of SBO-LSVRG under FL setting could be of vital importance.

## 7 CONCLUSION

In summary, this paper tackles the intricate challenges posed by Stochastic Bilevel Optimization (StocBO) within the realm of machine learning. We introduce the innovative SBO-LSVRG method, drawing inspiration from Loopless-SVRG (LSVRG). Remarkably, while retaining the optimal iteration complexity of the recently proposed SABA, our method markedly trims memory costs, presenting a robust solution for extensive problems. A pivotal highlight is that SBO-LSVRG represents the pioneering application of LSVRG in bilevel optimization, even under non-convex frameworks, and exhibits unparalleled convergence rates. These breakthroughs chart an exciting course for forthcoming inquiries and advancements in StocBO.

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

## CONTENTS

# A  OVERVIEW AND COMPARATIVE ANALYSIS OF VARIANCE REDUCTION TECHNIQUES

This section provides an overview and comparison of several pertinent variance reduction methods. We presuppose, for the sake of simplicity and without loss of generality, that our objective is to optimize a single-level finite-sum optimization model given by

$$\min_{x \in \mathbb{R}^d} F(x) = \frac{1}{m} \sum_{i=1}^{m} F_i(x),$$

where $x$ denotes the optimization parameter, $m$ represents the quantity of nodes, clients, or data points, and $F_i$ serves as the objective function for each index $i$ within the range of $[m]$. The focal point of our discussion revolves around the comparison of variance-reduced techniques including STOchastic Recursive Momentum (STORM) (Cutkosky & Orabona, 2019), Stochastic Average Gradient Algorithm (SAGA) (Defazio et al., 2014), StochAstic Recursive grAdient algoritHm (SARAH), and Loopless Stochastic Variance Reduced Gradient (L-SVRG) (Kovalev et al., 2020).

**STORM.**  STORM achieves variance reduction by utilizing a variant of the momentum term, akin to the momentum heuristic in the Adam optimization algorithm (Kingma & Ba, 2015). The momentum term is updated by the following rule:

$$c^t = g(x^t) + (1 - \gamma^t)(c^{t-1} - g(x^{t-1})),$$

given the decay parameter $\gamma^t$, the current gradient estimate $\nabla g(x^t)$, and the preceding gradient estimate $\nabla g(x^{t-1})$. Consequently, it is necessary to maintain the prior gradient $g(x^{t-1})$ and the previous momentum term $c^{t-1}$.

**SAGA.**  SAGA (Defazio et al., 2014) employs a gradient estimator for its operations. Assuming that we randomly sample the index $i$ from the range $[m]$, the gradient estimator for SAGA at iteration $t$ is given by:

$$g^t = \nabla f_i(x^t) - \nabla f_i(w_i^t) + \frac{1}{n} \sum_{j=1}^{n} \nabla f_j(w_j^t),$$

where $w_j^t$ denotes the stored local vector for each node $j$ and is updated as:

$$w_j^{t+1} = \begin{cases} x^t & j = i \\ w_j^t & j \neq i. \end{cases}$$

Despite its simplicity and impressive performance in practical scenarios, SAGA lacks memory efficiency due to the requirement to store $m$ local vectors, each with a dimensionality of $d$.

**SARAH.**  SARAH Nguyen et al. (2017) utilizes a recursive framework to update stochastic gradient estimates. Assuming $i$ is sampled uniformly at random from the set $[m]$, we define the gradient estimator as follows:

$$g^t = \nabla f_i(x^t) - \nabla f_i(x^{t-1}) + g^{t-1}, \quad \text{where} \quad x^{t+1} = x^t - \gamma g^t.$$

**L-SVRG.**  L-SVRG (Kovalev et al., 2020), a technique inspired by the original SVRG method (Johnson & Zhang, 2013), eliminates the outer loop and employs a probabilistic update of the full gradient. Assuming the index $i$ is sampled uniformly at random from the range $[m]$, the gradient estimator for L-SVRG at iteration $t$ is:

$$g^t = \nabla f_i(x^t) - \nabla f_i(w^t) + \nabla f(w^t),$$

with the globally stored weight $w^t$ updated as follows:

$$w^{t+1} = \begin{cases} x^t & \text{with probability } p \\ w^t & \text{with probability } 1 - p. \end{cases}$$

L-SVRG offers enhanced flexibility and memory efficiency when compared to SAGA. Firstly, we can select different values of $p$; when $p = 1/m$ (uniform sampling), Kovalev et al. (2020) recovers the same convergence rate as SAGA under convex settings. The performance under non-convex conditions remains uncertain; nonetheless, we present the first convergence result with $\mathcal{O}(1/\epsilon)$ in this paper. Secondly, we assert that L-SVRG is more efficient since it only requires storing one global weight $w^t$ rather than individual node-wise ones, a factor that inspired this paper.

# B  BASIC FACTS

## B.1  BREGMAN DIVERGENCE, $L$-SMOOTHNESS AND $\mu$-STRONGLY CONVEXITY

The Bregman divergence of a differentiable function $f : \mathbb{R}^d \to \mathbb{R}$ for all $x, y \in \mathbb{R}^d$ is defined by

$$D_f(x, y) := f(x) - f(y) - \langle \nabla f(y), x - y \rangle. \tag{8}$$

Then it is easy to have

$$\langle \nabla f(x) - \nabla f(y), x - y \rangle = D_f(x, y) + D_f(y, x), \quad \forall x, y \in \mathbb{R}^d. \tag{9}$$

We say a function $f : \mathbb{R}^d \to \mathbb{R}$ is $L$-smooth if for all $x, y \in \mathbb{R}^d$, we have

$$D_f(x, y) \le \frac{L}{2} \|x - y\|^2 \quad \text{and} \quad \frac{1}{2L} \|\nabla f(x) - \nabla f(y)\|^2 \le D_f(x, y). \tag{10}$$

Similarly, we say $f$ $\mu$-strongly convex if

$$\frac{\mu}{2} \|x - y\|^2 \le D_f(x, y) \quad \text{and} \quad D_f(x, y) \le \frac{1}{2\mu} \|\nabla f(x) - \nabla f(y)\|^2. \tag{11}$$

## B.2  VARIANCE DECOMPOSITIOIN

For a random vector $x \in \mathbb{R}^d$ (with finite second moment) and any $c \in \mathbb{R}^d$, the variance can be decomposed as

$$\mathbb{E}\left[\|x - \mathbb{E}[x]\|^2\right] = \mathbb{E}\left[\|x - c\|^2\right] - \|\mathbb{E}[x] - c\|^2. \tag{12}$$

## B.3  YOUNG'S INEQUALITY

For any two vectors $x, y \in \mathbb{R}^d$, we have

$$\|x + y\|^2 \le 2\|x\|^2 + 2\|y\|^2. \tag{13}$$

An extended version of Youngs's inequality is for all $x, y \in \mathbb{R}^d, \alpha > 0$, we have

$$\langle x, y \rangle \le \frac{\|x\|^2}{2\alpha} + \frac{\alpha \|y\|^2}{2}. \tag{14}$$

### B.4 JENSEN'S INEQUALITY

For a convex function $h : \mathbb{R}^d \to \mathbb{R}$ and any vectors $x_1, \cdots, x_n \in \mathbb{R}^d$, we have

$$h\left(\frac{1}{n}\sum_{i=1}^{n} x_i\right) \le \frac{1}{n}\sum_{i=1}^{n} h(x_i). \tag{15}$$

Consider the squared norm with $h(x) = \|x\|^2$, we have

$$\left\|\frac{1}{n}\sum_{i=1}^{n} x_i\right\|^2 \le \frac{1}{n}\sum_{i=1}^{n} \|x_i\|^2. \tag{16}$$

## C MISSING PROOFS

### C.1 PROOF OF THEOREM 1

**STEP 1: Bound the error between the iterates and the memories** We first control the error between the iterates and the memories. We define the following to make things simpler

$$E_y^t = \frac{1}{n}\sum_{i=1}^{n} \mathbb{E}\left[\left\|y^t - w_y^t\right\|^2\right], E_v^t = \frac{1}{n}\sum_{i=1}^{n} \mathbb{E}\left[\left\|v^t - w_v^t\right\|^2\right], E_x^t = \frac{1}{n}\sum_{i=1}^{n} \mathbb{E}\left[\left\|x^t - w_x^t\right\|^2\right]$$

For the corresponding calls to $F$, similarly we have $E_y'^t, E_v'^t$ and $E_x'^t$ with probabilities $q_x$ and $q_x'$.

We first prove the bound for $E_y^t$. Recall the updating rule of SBO-LSVRG,

$$w_y^{t+1} = \begin{cases} y^t & \text{with } q_y \\ w_y^t & \text{with } 1 - q_y, \end{cases}$$

we have

$$\mathbb{E}_t\left[\left\|y^{t+1} - w_y^{t+1}\right\|^2\right] = q_y \mathbb{E}_t\left[\left\|y^{t+1} - y^t\right\|^2\right] + (1 - q_y)\underbrace{\mathbb{E}_t\left[\left\|y^{t+1} - w_y^t\right\|^2\right]}_{T1} \tag{17}$$

Using the unbiasedness property of $\mathbb{E}_t\left[D_y^t(y^t, v^t, x^t)\right]$ in Eqn. 7, assume the constant learning rates $\rho$ and $\gamma$, we have

$$\begin{aligned} T1 &= \mathbb{E}_t\left[\left\|y^{t+1} - y^t + y^t - w_y^t\right\|^2\right] \\ &= \mathbb{E}_t\left[\left\|y^{t+1} - y^t\right\|^2\right] + \mathbb{E}_t\left[\left\|y^t - w_y^t\right\|^2\right] + 2\mathbb{E}\left[\langle y^{t+1} - y^t, y^t - w_y^t\rangle\right] \\ &\overset{7}{=} \mathbb{E}_t\left[\left\|y^{t+1} - y^t\right\|^2\right] + \mathbb{E}_t\left[\left\|y^t - w_y^t\right\|^2\right] - 2\rho\langle D_y(y^t, v^t, x^t), y^t - w_y^t\rangle \\ &\overset{16}{=} \mathbb{E}_t\left[\left\|y^{t+1} - y^t\right\|^2\right] + \frac{\rho}{\alpha}\left\|D_y(y^t, v^t, x^t)\right\|^2 + (1 + \rho\alpha)\left\|y^t - w_y^t\right\|^2 \end{aligned} \tag{18}$$

Put Eqn.18 into Eqn.17 and take the expectation over all $i$, we have

$$\begin{aligned} E_y^{t+1} &:= \frac{1}{n}\sum_{i=1}^{n} \mathbb{E}\left[\left\|y^{t+1} - w_y^{t+1}\right\|^2\right] \\ &\le (1 + \rho\alpha)(1 - q_y)E_y^t + \rho^2\mathbb{E}\left[\left\|D_y^t\right\|^2\right] + \frac{\rho}{\alpha}(1 - q_y)\mathbb{E}\left[\left\|D_y(y^t, v^t, x^t)\right\|^2\right]. \end{aligned} \tag{19}$$

By setting $\alpha = \frac{1}{2n\rho}$ and define $D_y(t) := D_y(y^t, v^t, x^t)$, we have

$$E_y^{t+1} \leq \left(1 - \frac{1}{2n}\right) E_y^t + \rho^2 \mathbb{E}\left[\left\|D_y^t\right\|^2\right] + 2n\rho^2 \mathbb{E}\left[\left\|D_y(t)\right\|^2\right], \tag{20}$$

Similarly, for other terms, we have

$$E_v^{t+1} \leq \left(1 - \frac{1}{2n}\right) E_v^t + \rho^2 \mathbb{E}\left[\left\|D_v^t\right\|^2\right] + 2n\rho^2 \mathbb{E}\left[\left\|D_v(t)\right\|^2\right],$$

$$E_x^{t+1} \leq \left(1 - \frac{1}{2n}\right) E_x^t + \gamma^2 \mathbb{E}\left[\left\|D_x^t\right\|^2\right] + 2n\rho^2 \mathbb{E}\left[\left\|D_x(t)\right\|^2\right],$$

$$E_y'^{t+1} \leq \left(1 - \frac{1}{2m}\right) E_y^t + \rho^2 \mathbb{E}\left[\left\|D_y^t\right\|^2\right] + 2n\rho^2 \mathbb{E}\left[\left\|D_y(t)\right\|^2\right], \tag{21}$$

$$E_v'^{t+1} \leq \left(1 - \frac{1}{2m}\right) E_v^t + \rho^2 \mathbb{E}\left[\left\|D_v^t\right\|^2\right] + 2n\rho^2 \mathbb{E}\left[\left\|D_v(t)\right\|^2\right],$$

$$E_x'^{t+1} \leq \left(1 - \frac{1}{2m}\right) E_x^t + \rho^2 \mathbb{E}\left[\left\|D_x^t\right\|^2\right] + 2n\gamma^2 \mathbb{E}\left[\left\|D_x(t)\right\|^2\right].$$

Noticed that for the above bounds, each term should not be necessary to be bounded with conditions $q_y, q_y', q_x, q_x'$.

**STEP 2: Bound the expected variances $D_y^t, D_v^t, D_x^t$.** Now we are ready to bound the variances w.r.t. $\mathbb{E}\left[\left\|D_y^t\right\|^2\right], \mathbb{E}\left[\left\|D_v^t\right\|^2\right], \mathbb{E}\left[\left\|D_x^t\right\|^2\right]$.

Suppose we consider $i$ sampled from $[n]$ at iteration $t$, we have
$$D_y^t = \nabla_y G_i(y^t, x^t) - \nabla_y G_i(w_y^t, w_x^t) + \nabla_y G(w_y^t, w_x^t) .$$

Then we have

$$\mathbb{E}_t[\|D_y^t(y^t, v^t, x^t)\|^2] = \mathbb{E}_t[\|\nabla_y G_i(y^t, x^t) - \nabla_y G_i(w_y^t, w_x^t) + \nabla_y G(w_y^t, w_x^t) \tag{22}$$

$$- \nabla_y G(y^t, x^t) + \nabla_y G(y^t, x^t)\|^2] \tag{23}$$

$$\overset{15}{\leq} 2\|\nabla_y G(y^t, x^t)\|^2 + 2\mathbb{E}_t[\|\nabla_y G_i(y^t, x^t) - \nabla_y G_i(w_y^t, w_x^t) \tag{24}$$

$$+ \nabla_y G(w_y^t, w_x^t) - \nabla_y G(y^t, x^t)\|^2] . \tag{25}$$

Using the unbiaseness property (Eqn. 7) again, the second term is the variance of $\nabla_y G_i(y^t, x^t) - \nabla_y G_i(w_y^t, w_x^t)$, which can be upper-bounded by

$$\mathbb{E}_t[\|\nabla_y G_i(y^t, x^t) - \nabla_y G_i(w_y^t, w_x^t) + \nabla_y G(w_y^t, w_x^t) - \nabla_y G(y^t, x^t)\|^2] \tag{26}$$

$$\overset{12}{=} \mathbb{E}_t[\|[\nabla_y G_i(y^t, x^t) - \nabla_y G_i(w_y^t, w_x^t)\|^2] - \left\|\mathbb{E}\left[\nabla_y G_i(y^t, x^t) - \nabla_y G_i(w_y^t, w_x^t)\right]\right\|^2 \tag{27}$$

$$\leq \mathbb{E}_t[\|[\nabla_y G_i(y^t, x^t) - \nabla_y G_i(w_y^t, w_x^t)\|^2] \tag{28}$$

$$= \frac{1}{n}\sum_{i=1}^{n} \|[\nabla_y G_i(y^t, x^t) - \nabla_y G_i(w_y^t, w_x^t)\|^2 \tag{29}$$

$$\overset{10}{\leq} \frac{L_y'}{n}\sum_{i=1}^{n}(\|y^t - w_y^t\|^2 + \|x^t - w_x^t\|^2) \tag{30}$$

where the last inequality comes from the Lipschitz continuity of each $\nabla_y G_i$ with $L_y' = \max_{i \in [n]} L_y^{G_i}$ with the definition of smoothness in Eqn. 10.

Then by putting equation 26 into equation 22 and taking the total expectation, we have

$$\mathbb{E}[\|D_y^t(y^t, v^t, x^t)\|]^2 \leq 2\mathbb{E}[\|D_y(y^t, v^t, x^t)\|^2] + 2L'_y(E_y^t + E_x^t) \ . \tag{31}$$

Later we consider the bound on $v$, it holds

$$\mathbb{E}_t[\|D_v^t(y^t, v^t, x^t)\|^2] = \mathbb{E}_t[\|\nabla_y F_j(y^t, x^t) - \nabla_y F_j(w_y'^t, w_x'^t) + \nabla_y F(w_y'^t, w_x'^t) \tag{32}$$

$$+ \nabla_{yy}^2 G_i(y^t, x^t)v^t - \nabla_{yy}^2 G_i(w_y^t, w_x^t)w_v^t + \nabla_{yy}^2 G(w_y'^t, w_x'^t)w_v'^t \tag{33}$$

$$- D_v(y^t, v^t, x^t) + D_v(y^t, v^t, x^t)\|^2] \tag{34}$$

$$\leq 2[\|D_v(y^t, v^t, x^t)\|^2 \tag{35}$$

$$+ 2\mathbb{E}_t[\|\nabla_y F_j(y^t, x^t) - \nabla_y F_j(w_y'^t, w_x'^t) + \nabla_y F(w_y'^t, w_x'^t) \tag{36}$$

$$+ \nabla_{yy}^2 G_i(y^t, x^t)v^t - \nabla_{yy}^2 G_i(w_y^t, w_x^t)w_v^t + \nabla_{yy} G^2(w_y'^t, w_x'^t)w_v'^t \tag{37}$$

$$- D_v(y^t, v^t, x^t)\|^2] \tag{38}$$

Then we control the variance of $\nabla_y F_j(y^t, x^t) - \nabla_y F_j(w_y'^t, w_x'^t) + \nabla_{yy}^2 G_i(y^t, x^t)v^t - \nabla_{yy}^2 G_i(w_y^t, w_x^t)w_v^t$. Since $i$ and $j$ are independent, this is a sum of two independent random variables, hence its variance is the sum of the variances, which is upper-bounded by

$$\mathbb{E}_t[\|\nabla_y F_j(y^t, x^t) - \nabla_y F_j(w_y'^t, w_x'^t)\|^2] + \mathbb{E}_t[\|\nabla_{yy}^2 G_i(y^t, x^t)v^t - \nabla_{yy}^2 G_i(w_y^t, w_x^t)w_v^t\|^2] \ .$$

For $\mathbb{E}_t[\|\nabla_y F_j(y^t, x^t) - \nabla_y F_j(w_y'^t, w_x'^t)\|^2]$ we use the Lipschitz continuity of the $\nabla_y F_j$:

$$\mathbb{E}_t[\|\nabla_y F_j(y^t, x^t) - \nabla_y F_j(w_y'^t, w_x'^t)\|^2] \leq \left[\max_{j \in [m]} L_1^{F_j}\right] \mathbb{E}_t[\|y^t - w_j^t\|^2 + \|x^t - w_x^t\|^2] \tag{39}$$

$$\leq \left[\max_{j \in [m]} L_1^{F_j}\right] \frac{1}{m} \sum_{j=1}^m (\|y^t - w_j^t\|^2 + \|x^t - w_x^t\|^2) \ . \tag{40}$$

Managing the expectation $\mathbb{E}_t[\|\nabla_{yy}^2 G_i(y^t, x^t)v^t - \nabla_{yy}^2 G_i(w_y^t, w_x^t)w_v^t\|^2]$ presents a greater challenge without the initial assumption of $v$'s boundedness. However, we can navigate around this complexity by introducing $\nabla_{yy}^2 G_i(y^*(x^t), x^t)v^*(x^t)$:

$$\mathbb{E}_t[\|\nabla_{yy}^2 G_i(y^t, x^t)v^t - \nabla_{yy}^2 G_i(w_y^t, w_x^t)w_v^t\|^2] \leq 4\{\mathbb{E}_t[\|\nabla_{yy}^2 G_i(y^t, x^t)(v^t - v^*(x^t))\|^2] \tag{41}$$

$$+ \mathbb{E}_t[\|(\nabla_{yy}^2 G_i(y^t, x^t) - \nabla_{yy}^2 G_i(y^*(x^t), x^t))v^*(x^t)\|^2]$$

$$+ \mathbb{E}_t[\|(\nabla_{yy}^2 G_i(y^*(x^t), x^t) - \nabla_{yy}^2 G_i(w_y^t, w_x^t))v^*(x^t)\|^2]$$

$$+ \mathbb{E}_t[\|\nabla_{yy}^2 G_i(w_y^t, w_x^t)(v^*(x^t) - w_v^t)\|^2]\}$$

$$\leq 4((\max_{i \in [n]} L_1^{G_i})\|v^t - v^*(x^t)\|^2 + (\max_{i \in [n]} L_2^{G_i})\frac{C^F}{\mu_G}\|y^t - y^*(x^t)\|^2 \tag{42}$$

$$+ (\max_{i \in [n]} L_2^{G_i})\frac{C^F}{\mu_G}(\|x^t - w_x^t\|^2 + 2(\|y^t - y^*(x^t)\|^2 + \|y^t - w_y^t\|^2))$$

$$+ (\max_{i \in [n]} L_1^{G_i})(\|x^t - w_x^t\|^2 + 2(\|v^t - v^*(x^t)\|^2 + \|v^t - w_v^t\|^2))$$

Taking the total expectation with defining $L'_v = 4\max\left(2\max_{i \in [n]} L_1^{G_i}, 2\max_{i \in [n]} L_2^{G_i}\frac{C^F}{\mu_G}, \max_{j \in [m]} L_1^{F_j}\right)$ and $L''_v = 4\max\left(3\max_{i \in [n]} L_1^{G_i}), 3\max_{i \in [n]} L_2^{G_i}\frac{C^F}{\mu_G}\right)$, we have

$$\mathbb{E}[\|D_v^t(y^t, v^t, x^t)\|^2] \leq 2\mathbb{E}[\|D_v(y^t, v^t, x^t)\|^2] + 2L'_v(E_y^t + E_x^t + E_v^t + E_z^t + E_x'^t) + 2L''_v(\delta_z^t + \delta_v^t). \tag{43}$$

Similarly for $x$ we have

$$\mathbb{E}[\|D_x^t(y^t, v^t, x^t)\|^2] \leq 2\mathbb{E}[\|D_x(y^t, v^t, x^t)\|^2] + 2L_x'(E_y^t + E_x^t + E_v^t + E_z'^t + E_x'^t) + 2L_x''(\delta_z^t + \delta_v^t) \tag{44}$$

We define $S^t = E_y^t + E_x^t + E_v^t + E_z'^t + E_v'^t + E_x'^t$, and letting $\Gamma = \min(\frac{1}{m}, \frac{1}{n})$. Note that by definition, each quantity $E_y^t$ is smaller than $S^t$. We will therefore use the cruder bounds on $\mathbb{E}[\|D_y^t\|^2]$, $\mathbb{E}[\|D_v^t\|^2]$ and $\mathbb{E}[\|D_x^t\|^2]$ as follows

$$\mathbb{E}[\|D_y^t(y^t, v^t, x^t)\|^2] \leq 2L_z^2 \delta_y^t + 2L_z' S^t \tag{45}$$

$$\mathbb{E}[\|D_v^t(y^t, v^t, x^t)\|^2] \leq 2(L_v^2 + L_v'')(\delta_y^t + \delta_v^t) + 2L_v' S^t \tag{46}$$

and

$$\mathbb{E}[\|D_x^t(y^t, v^t, x^t)\|^2] \leq 2\mathbb{E}[\|D_x\|^2] + 2L_x' S^t + 2L_x''(\delta_z^t + \delta_v^t) \ . \tag{47}$$

We have the following lemma

**Lemma 3** *If* $4\rho^2(L_z' + L_v') + 4\gamma^2 L_x' \leq \frac{\Gamma}{2}$ *and* $4L_x'' \gamma^2 \leq \rho^2(L_v^2 + 4L_v'')$, *it holds*

$$S^{t+1} \leq \left(1 - \frac{\Gamma}{2}\right) S^t + \beta_{sz} \rho^2 \delta_y^t + \beta_{sv} \rho^2 \delta_v^t + P\gamma^2 \mathbb{E}[\|D_x\|^2]$$

*for some* $L_s, \beta_{sz}, P > 0$.

*Proof:* It holds based on the previous analysis

$$S^{t+1} \leq (1 - \Gamma) S^t + \mathbb{E}\left[2\rho^2(\|D_y^t\|^2 + \|D_v^t\|^2) + 2\gamma^2 \|D_x^t\|^2\right.$$
$$\left. + 2(m+n)[\rho^2(\|D_z\|^2 + \|D_v\|^2) + \gamma^2 \|D_x\|^2]\right] \ .$$

Using the previous bounds equation 31, equation 43 and equation 44, we get

$$S^{t+1} \leq \left(1 - \Gamma + 4\rho^2(L_z' + L_v') + 4\gamma^2 L_x'\right) S^t + (2(m+n) + 4)\mathbb{E}[\rho^2(\|D_z\|^2 + \|D_v\|^2)$$
$$+ \gamma^2 \|D_x\|^2] + 4L_v'' \rho^2(\delta_z^t + \delta_v^t) + 4L_x'' \gamma^2(\delta_z^t + \delta_v^t) \ .$$

Next, using $4\rho^2(L_z' + L_v') + 4\gamma^2 L_x' \leq \frac{\Gamma}{2}$ and letting $P = (2(m+n) + 4)$ we get

$$S^{t+1} \leq \left(1 - \frac{\Gamma}{2}\right) S^t + P\mathbb{E}[\rho^2(\|D_z\|^2 + \|D_v\|^2) + \gamma^2 \|D_x\|^2] + +4L_v'' \rho^2(\delta_z^t + \delta_v^t) + 4L_x'' \gamma^2(\delta_z^t + \delta_v^t).$$

To finish, we use 1 to have

$$S^{t+1} \leq \left(1 - \frac{\Gamma}{2}\right) S^t + P[\rho^2((L_y^2 + L_v^2)\delta_y^t + L_v^2 \delta_v^t) + (4L_v'' \rho^2 + 4L_x'' \gamma^2)(\delta_z^t + \delta_v^t) + \gamma^2 \mathbb{E}[\|D_x\|^2]].$$

Then, using that $4L_x'' \gamma^2 \leq \rho^2(L_v^2 + 4L_v'')$, we get the bound, letting $L_{sz} = L_y^2 + L_v^2 + 4L_v''$ and $L_{sv} = L_v^2 + 4L_v''$:

$$S^{t+1} \leq \left(1 - \frac{\Gamma}{2}\right) S^t + \beta_{sz} \rho^2 \delta_y^t + \beta_{sv} \rho^2 \delta_v^t + P\gamma^2 \mathbb{E}[\|D_x\|^2]$$

with $\beta_{sz} = 2PL_{sz}$, $\beta_{sv} = 2PL_{sv}$. ∎

**STEP 3: Putting it all together.** Recall that we denote $g^t = \mathbb{E}[\|\nabla h(x^t)\|^2]$ and $h^t = \mathbb{E}[h(x^t)]$, hence we have the following descent lemma

**Lemma 4** *If*

$$\rho \leq \min\left(\frac{\mu_G}{64L_y^2}, \frac{\overline{\beta}_{zx}}{2\beta_{zx}}, \frac{\mu_G}{128(L_v^2 + L_v'')}, \frac{\beta_{vz}}{8(L_v^2 + L_v'')}, \frac{\overline{\beta}_{vx}}{2\beta_{vx}}\right)$$

*and*

$$\gamma \leq \min\left(\sqrt{\frac{\rho\mu_G}{64\beta_{zx}L_x''}}, \sqrt{\frac{L_z'}{2L_x'\beta_{zx}}}\rho, \sqrt{\frac{\rho\mu_G}{128\beta_{vx}L_x''}}, \sqrt{\frac{\rho\beta_{vz}}{4L_x''\beta_{vx}}}, \sqrt{\frac{L_v'}{2L_x'\beta_{vx}}}\rho, \frac{1}{4L^h}, \frac{L_x^2}{2L^hL_x''}\right)$$

*then it holds*

$$\delta_z^{t+1} \leq \left(1 - \frac{\rho\mu_G}{8}\right)\delta_z^t + 2L_x''\beta_{zx}\gamma^2\delta_v^t + 5L_z'\rho^2 S^t + 2\overline{\beta}_{zx}\frac{\gamma^2}{\rho}\mathbb{E}[\|D_x(y^t, v^t, x^t)\|^2] \ , \quad (48)$$

$$\delta_v^{t+1} \leq \left(1 - \frac{\rho\mu_G}{16}\right)\delta_v^t + 3\beta_{vz}\rho\delta_y^t + 5L_v'\rho^2 S^t + 2\overline{\beta}_{vx}\frac{\gamma^2}{\rho}\mathbb{E}[\|D_x(y^t, v^t, x^t)\|^2] \quad (49)$$

*and*

$$h^{t+1} \leq h^t - \frac{\gamma}{2}g^t - \frac{\gamma}{4}\mathbb{E}[\|D_x(y^t, v^t, x^t)\|^2] + L_x^2\gamma(\delta_z^t + \delta_v^t) + L^hL_x'\gamma^2 S^t \ . \quad (50)$$

*Proof:* We first have

$$\delta_z^{t+1} \leq \left(1 - \frac{\rho\mu_G}{4} + 4L_y^2\rho^2 + 4\beta_{zx}L_x''\gamma^2\right)\delta_z^t + 2L_x''\beta_{zx}\gamma^2\delta_v^t \quad (51)$$
$$+ (4L_z'\rho^2 + 2L_x'\beta_{zx}\gamma^2)S^t + \left(2\beta_{zx}\gamma^2 + \overline{\beta}_{zx}\frac{\gamma^2}{\rho}\right)\mathbb{E}[\|D_x(y^t, v^t, x^t)\|^2]$$

Since $\rho \leq \frac{\mu_G}{64L_y^2}$ and $\gamma^2 \leq \frac{\rho\mu_G}{64\beta_{zx}L_x''}$, we have

$$-\frac{\rho\mu_G}{4} + 4L_y^2\rho^2 + 4\beta_{zx}L_x''\gamma^2 \leq -\frac{\rho\mu_G}{8} \ . \quad (52)$$

The condition $\gamma^2 \leq \frac{L_z'}{2L_x'\beta_{zx}}\rho^2$ gives us

$$4L_z'\rho^2 + 2L_x'\beta_{zx}\gamma^2 \leq 5L_z'\rho^2 \ . \quad (53)$$

With $\rho \leq \frac{\overline{\beta}_{zx}}{2\beta_{zx}}$, we get

$$2\beta_{zx}\gamma^2 + \overline{\beta}_{zx}\frac{\gamma^2}{\rho} \leq 2\overline{\beta}_{zx}\frac{\gamma^2}{\rho} \ . \quad (54)$$

We can plug Equations equation 52, equation 53 and equation 54 into above and we end up with

$$\delta_z^{t+1} \leq \left(1 - \frac{\rho\mu_G}{8}\right)\delta_z^t + 2L_x''\beta_{zx}\gamma^2\delta_v^t + 5L_z'\rho^2 S^t + 2\overline{\beta}_{zx}\frac{\gamma^2}{\rho}\mathbb{E}[\|D_x(y^t, v^t, x^t)\|^2] \ .$$

The proof for $\delta_v^t$ is quite similar. From above,

$$\delta_v^{t+1} \leq \left(1 - \frac{\rho\mu_G}{8}\right)\delta_v^t + \beta_{vz}\rho\delta_y^t + 2\rho^2 V_v^t + \beta_{vx}\gamma^2 V_x^t + \overline{\beta}_{vx}\frac{\gamma^2}{\rho}\mathbb{E}[\|D_x(y^t, v^t, x^t)\|^2] \quad (55)$$

$$\leq \left(1 - \frac{\rho\mu_G}{8} + 4(L_v^2 + L_v'')\rho^2 + 4L_x''\beta_{vx}\gamma^2\right)\delta_v^t + (4(L_v^2 + L_v'')\rho^2 + 2L_x''\beta_{vx}\gamma^2 + \beta_{vz}\rho)\delta_y^t + \quad (56)$$

$$+ \left(4L_v'\rho^2 + 2L_x'\beta_{vx}\gamma^2\right)S^t + \left(2\beta_{vx}\gamma^2 + \overline{\beta}_{vx}\frac{\gamma^2}{\rho}\right)\mathbb{E}[\|D_x(y^t, v^t, x^t)\|^2] \ .$$

Using $\rho \leq \frac{\mu_G}{128(L_v^2 + L_v'')}$ and $\gamma^2 \leq \frac{\rho\mu_G}{128L_x''\beta_{vx}}$, we get

$$-\frac{\rho\mu_G}{8} + 4(L_v^2 + L_v'')\rho^2 + 4L_x''\beta_{vx}\gamma^2 \leq -\frac{\rho\mu_G}{16} \ . \quad (57)$$

With $\gamma^2 \leq \frac{\rho \beta_{vz}}{4L_x'' \beta_{vx}}$ and $\rho \leq \frac{\beta_{vz}}{8(L_v^2 + L_v'')}$, we have

$$4(L_v^2 + L_v'')\rho^2 + 2L_x'' \beta_{vx} \gamma^2 + \beta_{vz}\rho \leq 3\beta_{vz}\rho \ . \tag{58}$$

The condition $\gamma^2 \leq \frac{L_v'}{2L_x' \beta_{vx}}\rho^2$ yields

$$4L_v'\rho^2 + 2L_x' \beta_{vx} \gamma^2 \leq 5L_v'\rho^2 \ . \tag{59}$$

With $\rho \leq \frac{\overline{\beta}_{vx}}{2\beta_{vx}}$ we get

$$2\beta_{vx}\gamma^2 + \overline{\beta}_{zx}\frac{\gamma^2}{\rho} \leq 2\overline{\beta}_{vx}\frac{\gamma^2}{\rho} \ . \tag{60}$$

As a consequence of Equations equation 57, equation 58, equation 59 and equation 60, we have

$$\delta_v^{t+1} \leq \left(1 - \frac{\rho \mu_G}{16}\right)\delta_v^t + 3\beta_{vz}\rho \delta_y^t + 5L_v'\rho^2 S^t + 2\overline{\beta}_{vx}\frac{\gamma^2}{\rho}\mathbb{E}[\|D_x(y^t, v^t, x^t)\|^2] \ .$$

For the inequality on $h^t$, we start from equation 47

$$h^{t+1} \leq h^t - \frac{\gamma}{2}g^t - \left(\frac{\gamma}{2} - L^h\gamma^2\right)\mathbb{E}[\|D_x(y^t, v^t, x^t)\|^2] \tag{61}$$
$$+ \left(\frac{L_x^2}{2}\gamma + L^h L_x''\gamma^2\right)(\delta_z^t + \delta_v^t) + L^h L_x'\gamma^2 S^t \ .$$

Assuming $\gamma \leq \min\left(\frac{1}{4L^h}, \frac{L_x^2}{2L^h L_x''}\right)$ leads

$$h^{t+1} \leq h^t - \frac{\gamma}{2}g^t - \frac{\gamma}{4}\mathbb{E}[\|D_x(y^t, v^t, x^t)\|^2] + L_x^2\gamma(\delta_z^t + \delta_v^t) + L^h L_x'\gamma^2 S^t \ . \tag{62}$$

■

We are now ready to prove the final results.

*Proof:* We consider the Lyapunov function

$$\mathcal{L}^t = h^t + \phi_s S^t + \phi_z \delta_y^t + \phi_v \delta_v^t \tag{63}$$

for some constants $\phi_s$, $\phi_z$ and $\phi_v$.

We have

$$\mathcal{L}^{t+1} - \mathcal{L}^t \leq -\frac{\gamma}{2}g^t - \left(\frac{\gamma}{4} - 2\phi_z \overline{\beta}_{zx}\frac{\gamma^2}{\rho} - 2\phi_v \overline{\beta}_{vx}\frac{\gamma^2}{\rho} - \phi_s P\gamma^2\right)\mathbb{E}[\|D_x(y^t, v^t, x^t)\|^2]$$
$$- \left(\phi_z \frac{\mu_G}{8}\rho - L_x^2\gamma - 8\phi_v \beta_{vz}\rho - \phi_s \beta_{sz}\rho^2\right)\delta_z^t$$
$$- \left(\phi_v \frac{\mu_G}{16}\rho - L_x^2\gamma - 2\phi_z L_x''\gamma^2 - \phi_s \beta_{sv}\rho^2\right)\delta_v^t$$
$$- \left(\phi_s \frac{\Gamma}{2} - 5\phi_z L_z'\rho^2 - 5\phi_v L_v'\rho^2 - L^h L_x'\gamma^2\right)S^t \ .$$

To get a decrease, $\phi_z$, $\phi_v$ and $\phi_s$, $\rho$ and $\gamma$ must be such that:

$$2\phi_z \overline{\beta}_{zx}\frac{\gamma^2}{\rho} + 2\phi_v \overline{\beta}_{vx}\frac{\gamma^2}{\rho} + \phi_s P\gamma^2 \leq \frac{\gamma}{4}$$

$$L_x^2\gamma + 8\phi_v \beta_{vz}\rho + \phi_s \beta_{sz}\rho^2 \leq \phi_z \frac{\mu_G}{8}\rho$$

$$L_x^2\gamma + 8\phi_z L_x''\gamma^2 + \phi_s \beta_{sv}\rho^2 \leq \phi_v \frac{\mu_G}{16}\rho$$

$$5\phi_z L_z'\rho^2 + 5\phi_v L_v'\rho^2 + L^h L_x'\gamma^2 \leq \phi_s \frac{\Gamma}{2} \ .$$

In order to take into account the scaling of the quantities with respect to $N = n + m$, we take $\rho = \rho' N^{n_\rho}$, $\gamma = \gamma' N^{n_\gamma}$, $\phi_z = \phi'_z N^{n_z}$, $\phi_v = \phi'_v N^{n_v}$ and $\phi_s = \phi'_s N^{n_s}$. Since $\Gamma = \mathcal{O}(N^{-1})$, $P = \mathcal{O}(N)$, $\beta_{sz} = \mathcal{O}(N)$ and $\beta_{sv} = \mathcal{O}(N)$, we also define $\Gamma' = \Gamma N$, $P' = P N^{-1}$, $\beta'_{sz} = \beta_{sz} N^{-1}$ and $\beta'_{sv} N^{-1}$. Now, the previous Equations read (after slight simplifications):

$$(2\phi'_z \overline{\beta}_{zx} + 2\phi'_v \overline{\beta}_{vx}) \frac{\gamma'}{\rho'} N^{n_z + n_\gamma - n_\rho} + \phi'_s P' \gamma' N^{n_s + n_\gamma + 1} \leq \frac{1}{4}$$

$$L_x^2 \gamma' N^{n_\gamma} + 8\phi'_v \beta_{vz} \rho' N^{n_v + n_\rho} + \phi'_s \beta'_{sz} (\rho')^2 N^{2n_\rho + n_s + 1} \leq \phi'_z \frac{\mu_G}{8} \rho' N^{n_z + n_\rho}$$

$$L_x^2 \gamma' N^{n_\gamma} + 8\phi'_z L''_x (\gamma')^2 N^{2n_\gamma + n_z} + \phi'_s \beta'_{sv} (\rho')^2 N^{n_s + 2n_\rho + 1} \leq \phi'_v \frac{\mu_G}{16} \rho' N^{n_v + n_\rho}$$

$$5\phi'_z L'_z (\rho')^2 N^{n_z + 2n_\rho} + 5\phi'_v L'_v (\rho')^2 N^{2n_\rho + n_v} + L^h L'_x (\gamma')^2 N^{n_\gamma} \leq \phi'_s \frac{\Gamma'}{2} N^{n_s - 1} \quad .$$

In order to ensure that the exponents on $N$ are lower in the left-hand-side than those on the right-hand-side, we take $n_z = n_v = 0$, $n_\rho = n_\gamma = -\frac{2}{3}$ and $n_s = -\frac{1}{3}$. The Equations become

$$(2\phi'_z \overline{\beta}_{zx} + 2\phi'_v \overline{\beta}_{vx}) \frac{\gamma'}{\rho'} + \phi'_s P' \gamma' \leq \frac{1}{4}$$

$$L_x^2 \gamma' N^{-2/3} + 8\phi'_v \beta_{vz} \rho' N^{-2/3} + \phi'_s \beta'_{sz} (\rho')^2 N^{-2/3} \leq \phi'_z \frac{\mu_G}{8} \rho' N^{-2/3}$$

$$L_x^2 \gamma' N^{-2/3} + 8\phi'_z L''_x (\gamma')^2 N^{-4/3} + \phi'_s \beta'_{sv} (\rho')^2 N^{-2/3} \leq \phi'_v \frac{\mu_G}{16} \rho' N^{-2/3}$$

$$5\phi'_z L'_z (\rho')^2 N^{-4/3} + 5\phi'_v L'_v (\rho')^2 N^{-4/3} + L^h L'_x (\gamma')^2 N^{-4/3} \leq \phi'_s \frac{\Gamma'}{2} N^{-4/3} \quad .$$

We can replace the penultimate equation by the stronger

$$L_x^2 \gamma' N^{-2/3} + 8\phi'_z L''_x (\gamma')^2 N^{-2/3} + \phi'_s \beta'_{sv} (\rho')^2 N^{-2/3} \leq \phi'_v \frac{\mu_G}{16} \rho' N^{-2/3}$$

so that we can simplify all the equations by dropping the dependencies in $N$:

$$(2\phi'_z \overline{\beta}_{zx} + 2\phi'_v \overline{\beta}_{vx}) \frac{\gamma'}{\rho'} + \phi'_s P' \gamma' \leq \frac{1}{4}$$

$$L_x^2 \gamma' + 8\phi'_v \beta_{vz} \rho' + \phi'_s \beta'_{sz} (\rho')^2 \leq \phi'_z \frac{\mu_G}{8} \rho'$$

$$L_x^2 \gamma' + 8\phi'_z L''_x (\gamma')^2 + \phi'_s \beta'_{sv} (\rho')^2 \leq \phi'_v \frac{\mu_G}{16} \rho'$$

$$5\phi'_z L'_z (\rho')^2 + 5\phi'_v L'_v (\rho')^2 + L^h L'_x (\gamma')^2 \leq \phi'_s \frac{\Gamma'}{2} \quad .$$

Let us take $\phi'_s = 1$, $\phi'_z = \phi''_z \frac{\rho'}{\gamma'}$ and $\phi'_v = \phi''_v \frac{\rho'}{\gamma'}$ with $\phi''_z = \frac{1}{32\overline{\beta}_{zx}}$ and $\phi''_v = \min\left(\frac{1}{32\overline{\beta}_{vx}}, \phi''_z \frac{\mu_G}{128\beta_{vz}}\right)$. The equations become

$$P' \gamma' \leq \frac{1}{8}$$

$$L_x^2 \gamma' + \beta'_{sz} (\rho')^2 \leq \phi''_z \frac{\mu_G}{16} \frac{(\rho')^2}{\gamma'}$$

$$L_x^2 \gamma' + 8\phi''_z L''_x \gamma' \rho' + \beta'_{sv} (\rho')^2 \leq \phi''_v \frac{\mu_G}{16} \frac{(\rho')^2}{\gamma'}$$

$$5\phi''_z L'_z \frac{(\rho')^3}{\gamma'} + 5\phi''_v L'_v \frac{(\rho')^3}{\gamma'} + L^h L'_x (\gamma')^2 \leq \frac{\Gamma'}{2} \quad .$$

The condition $\gamma' \leq \frac{1}{8P'}$ ensures that the first equation is verified. With $\gamma' \leq \min\left(\sqrt{\frac{\phi''_z \mu_G}{32 L_x^2}} \rho', \frac{\phi''_z \mu_G}{32\beta'_{sz}}\right)$, the second equations is verified. With $\gamma' \leq \min\left(\sqrt{\frac{\phi''_v \mu_G}{48 L_x^2}} \rho', \frac{\phi''_v \mu_G}{48\beta'_{sv}}, \sqrt{\frac{\phi''_v \mu_G}{384\phi''_z L''_x \rho'}}\right)$, the third is verified. With $\gamma' \leq \sqrt{\frac{\Gamma'}{6L^h L'_x}}$, the last can be simplified:

$$(5\phi''_z L'_z + 5\phi''_v L'_v)(\rho')^3 \leq \frac{\Gamma'}{3} \gamma' \quad .$$

Let us write $\gamma' = \xi\rho'$. If we want that equation does no contradict the previous upper bound on $\gamma'$ involving $\rho'$ and the conditions of lemma 4, that is

$$\gamma' \leq \underbrace{\min\left(\sqrt{\frac{\phi_z''\mu_G}{32L_x^2}}, \sqrt{\frac{\phi_v''\mu_G}{48L_x^2}}, \sqrt{\frac{L_z'}{2L_x'\beta_{zx}}}, \sqrt{\frac{L_v'}{2L_x'\beta_{vx}}}\right)}_{K_1} \rho'$$

$$\gamma' \leq \underbrace{\min\left(\sqrt{\frac{\mu_G}{64\beta_{zx}L_x''}}, \sqrt{\frac{\mu_G}{128\beta_{vx}L_x''}}, \sqrt{\frac{\beta_{vz}}{4L_x''\beta_{vx}}}\right)}_{K_2} \sqrt{\rho'}$$

$$\gamma' \leq \underbrace{\sqrt{\frac{\phi_v''\mu_G}{384\phi_z''L_x''}}}_{K_3} \frac{1}{\sqrt{\rho'}}$$

$$\gamma' \leq \underbrace{\min\left(\frac{1}{4L^h}, \frac{L_x^2}{2L^h L_x''}, \sqrt{\frac{\Gamma'}{6L^h L_x'}}, \frac{1}{8P'}, \frac{\phi_z''\mu_G}{32\beta_{sz}'}, \frac{\phi_v''\mu_G}{48\beta_{sv}'}\right)}_{K_4}$$

$$\gamma' \geq \underbrace{\frac{15(\phi_z''L_z' + \phi_v''L_v')}{\Gamma'}}_{K_5} \rho^3$$

$\xi$ must verify

$$\xi \leq K_1$$
$$\xi \leq K_2(\rho')^{-\frac{1}{2}}$$
$$\xi \leq K_3(\rho')^{-\frac{3}{2}}$$
$$\xi \leq K_4(\rho')^{-1}$$
$$\xi \geq K_5(\rho')^2$$

which is possible if $\rho'$ satisfies

$$\rho' \leq \min\left(\sqrt{\frac{K_1}{K_5}}, \left(\frac{K_2}{K_5}\right)^{-\frac{3}{2}}, \left(\frac{K_3}{K_5}\right)^{-\frac{5}{2}}, \left(\frac{K_4}{K_5}\right)^{-2}\right) .$$

Let us take

$$\rho' = \min\left(\sqrt{\frac{K_1}{K_5}}, \left(\frac{K_2}{K_5}\right)^{-\frac{3}{2}}, \left(\frac{K_3}{K_5}\right)^{-\frac{5}{2}}, \left(\frac{K_4}{K_5}\right)^{-2}, \frac{\mu_G}{64L_y^2}, \frac{\overline{\beta}_{zx}}{2\beta_{zx}}, \frac{\mu_G}{128(L_v^2 + L_v'')}, \frac{\beta_{vz}}{8(L_v^2 + L_v'')}, \frac{\overline{\beta}_{vx}}{2\beta_{vx}}\right) \tag{64}$$

and

$$\xi = \min(K_1, K_2(\rho')^{-\frac{1}{2}}, K_3(\rho')^{-\frac{3}{2}}, K_4(\rho')^{-1}) . \tag{65}$$

Finally, we have

$$\mathcal{L}^{t+1} - \mathcal{L}^t \leq -\frac{\gamma}{2}g^t$$

and therefore, summing and telescoping yields

$$\frac{1}{T}\sum_{t=1}^{T} g^t \leq \frac{\mathcal{L}^1}{\gamma T} = \frac{\mathcal{L}^0 N^{\frac{2}{3}}}{T} .$$

Since with respect to $N$ we have

$$\mathcal{L}^0 = h^0 + \phi_z\delta_z^0 + \phi_v\delta_v^0 + \phi_s S^0 = \mathcal{O}(N^{-1} + 1 + 1 + N^{-\frac{1}{3}}) = \mathcal{O}(1) ,$$

we end up with

$$\frac{1}{T}\sum_{t=1}^{T}\mathbb{E}[\|\nabla h(x^t)\|^2] = \mathcal{O}\left(\frac{N^{\frac{2}{3}}}{T}\right) \ .$$

∎

## C.2  PROOF OF THEOREM 2

Here, we have

$$\rho' = \min\left(\sqrt{\frac{K_1'}{K_5}}, \left(\frac{K_2}{K_5}\right)^{\frac{2}{5}}, \left(\frac{K_3}{K_5}\right)^{\frac{2}{7}}, \left(\frac{K_4'}{K_5}\right)^{\frac{1}{3}}, \frac{\mu_G}{64L_z^2}, \frac{\overline{\beta}_{zx}}{2\beta_{zx}}, \frac{\mu_G}{128(L_v^2 + L_v'')}, \frac{\beta_{vz}}{8(L_v^2 + L_v'')}, \frac{\overline{\beta}_{vx}}{2\beta_{vx}}\right) \ ,$$

and

$$\xi = \min(K_1', K_2(\rho')^{-\frac{1}{2}}, K_3(\rho')^{-\frac{3}{2}}, K_4'(\rho')^{-1}) \ .$$

where $P' = PN^{-1}, \Gamma' = \Gamma N$,

$$\phi_z'' = \frac{1}{32\overline{\beta}_{zx}} \ , \phi_v'' = \min\left(\frac{1}{32\overline{\beta}_{vx}}, \phi_z''\frac{\mu_G}{128\beta_{vz}}\right) \ ,$$

$$K_1' = \min\left(\frac{\mu_G}{64c'}, \sqrt{\frac{\phi_z''\mu_G}{48L_x^2}}, \sqrt{\frac{\phi_v''\mu_G}{64L_x^2}}, \sqrt{\frac{L_z'}{2L_x'\beta_{zx}}}, \sqrt{\frac{L_v'}{2L_x'\beta_{vx}}}\right) \ ,$$

$$K_2 = \min\left(\sqrt{\frac{\mu_G}{64\beta_{zx}L_x''}}, \sqrt{\frac{\mu_G}{128\beta_{vx}L_x''}}, \sqrt{\frac{\beta_{vz}}{4L_x''\beta_{vx}}}\right) \ ,$$

$$K_3 = \sqrt{\frac{\phi_v''\mu_G}{512\phi_z''L_x''}} \ , \quad K_4' = \min\left(\frac{\Gamma'}{6c'}, \frac{1}{4L^\Phi}, \frac{L_x^2}{2L^\Phi L_x''}, \sqrt{\frac{\Gamma'}{6L^\Phi L_x'}}, \frac{1}{18P'}, \frac{\phi_z''\mu_G}{48\beta_{sz}'}, \frac{\phi_v''\mu_G}{64\beta_{sv}'}\right)$$

and

$$K_5 = \frac{20(\phi_z''L_z' + \phi_v''L_v')}{\Gamma'} \ .$$

*Proof:*

For simplicity, we assume that $\Phi^* = 0$ and so for any $x \in \mathbb{R}^d$ the PL inequality reads:

$$\frac{1}{2}\|\nabla\Phi(x)\|^2 \geq \mu_\Phi\Phi(x) \ . \tag{66}$$

Then, eq. equation 50 gives

$$\Phi^{t+1} \leq \left(1 - \frac{\gamma\mu_\Phi}{2}\right)\Phi^t - \frac{\gamma}{4}\mathbb{E}[\|D_x(z^t, v^t, x^t)\|^2] + \gamma L_x^2(\delta_z^t + \delta_v^t) + L^\Phi L_x'\gamma^2 S^t \ .$$

We take $\mathcal{L}^t$ the Lyapunov function given in eq. (63). We find

$$\mathcal{L}^{t+1} - \mathcal{L}^t \leq -\gamma\mu_\Phi\Phi^t - \left(\frac{\gamma}{4} - 2\phi_z\overline{\beta}_{zx}\frac{\gamma^2}{\rho} - 2\phi_v\overline{\beta}_{vx}\frac{\gamma^2}{\rho} - \phi_s P\gamma^2\right)\mathbb{E}[\|D_x(z^t, v^t, x^t)\|^2]$$

$$- \left(\phi_z\frac{\mu_G}{8}\rho - L_x^2\gamma - 8\phi_v\beta_{vz}\rho - \phi_s\beta_{sz}\rho^2\right)\delta_z^t$$

$$- \left(\phi_v\frac{\mu_G}{16}\rho - L_x^2\gamma - 2\phi_z L_x''\gamma^2 - \phi_s\beta_{sv}\rho^2\right)\delta_v^t$$

$$- \left(\phi_s\frac{\Gamma}{2} - 5\phi_z L_z'\rho^2 - 5\phi_v L_v'\rho^2 - L^\Phi L_x'\gamma^2\right)S^t \ .$$

We now try to find linear convergence, hence we add to this $c\mathcal{L}^t$ to get

$$\mathcal{L}^{t+1} - (1-c)\mathcal{L}^t \leq -(\gamma\mu_\Phi - c)\Phi^t - \left(\frac{\gamma}{4} - 2\phi_z\overline{\beta}_{zx}\frac{\gamma^2}{\rho} - 2\phi_v\overline{\beta}_{vx}\frac{\gamma^2}{\rho} - \phi_s P\gamma^2 - c\right)\mathbb{E}[\|D_x(z^t, v^t, x^t)\|^2]$$

$$- \left(\phi_z\frac{\mu_G}{8}\rho - L_x^2\gamma - 8\phi_v\beta_{vz}\rho - \phi_s\beta_{sz}\rho^2 - c\phi_z\right)\delta_z^t$$

$$- \left(\phi_v\frac{\mu_G}{16}\rho - L_x^2\gamma - 2\phi_z L_x''\gamma^2 - \phi_s\beta_{sv}\rho^2 - c\phi_v\right)\delta_v^t$$

$$- \left(\phi_s\frac{\Gamma}{2} - 5\phi_z L_z'\rho^2 - 5\phi_v L_v'\rho^2 - L^\Phi L_x'\gamma^2 - c\phi_S\right)S^t \ .$$

Hence, the set of inequations for decrease becomes

$$c \leq \gamma\mu_\Phi$$

$$2\phi_z\overline{\beta}_{zx}\frac{\gamma^2}{\rho} + 2\phi_v\overline{\beta}_{vx}\frac{\gamma^2}{\rho} + \phi_s P\gamma^2 + c \leq \frac{\gamma}{4}$$

$$L_x^2\gamma + 8\phi_v\beta_{vz}\rho + \phi_s\beta_{sz}\rho^2 + \phi_z c \leq \phi_z\frac{\mu_G}{8}\rho$$

$$L_x^2\gamma + 8\phi_z L_x''\gamma^2 + \phi_s\beta_{sv}\rho^2 + \phi_v c \leq \phi_v\frac{\mu_G}{16}\rho$$

$$5\phi_z L_z'\rho^2 + 5\phi_v L_v'\rho^2 + L^\Phi L_x'\gamma^2 + \phi_s c \leq \phi_s\frac{\Gamma}{2} \ .$$

We see that it is more convenient to write $c = \gamma c'$. As previously, we write $\gamma = \gamma' N^{n_\gamma}$, $\rho = \rho' N^{n_\rho}$, $\phi_z = \phi_z' N^{n_z}$, $\phi_v = \phi_v' N^{n_v}$, $\phi_s = \phi_s' N^{n_s}$, $P = P'N$, $\Gamma = \Gamma' N^{-1}$, $\beta_{sx} = \beta_{sx}' N$ and $\beta_{sv} = \beta_{sv}' N$. The equations read:

$$c' \leq \mu_\Phi$$

$$2\phi_z'\overline{\beta}_{zx}\frac{\gamma'}{\rho'}N^{n_z+n_\gamma-n_\rho} + 2\phi_v'\overline{\beta}_{vx}\frac{\gamma'}{\rho'}N^{n_v+n_\gamma-n_\rho} + \phi_s'P'\gamma'N^{n_s+1+n_\gamma} + c' \leq \frac{1}{4}$$

$$L_x^2\gamma'N^{n_\gamma} + 8\phi_v'\beta_{vz}\rho'N^{n_v+n_\rho} + \phi_s'\beta_{sz}'(\rho')^2 N^{n_s+2n_\rho+1} + \phi_z'c'\gamma'N^{n_z+n_\gamma} \leq \phi_z'\frac{\mu_G}{8}\rho'N^{n_\rho+n_z}$$

$$L_x^2\gamma'N^{n_\gamma} + 8\phi_z'L_x''(\gamma')^2 N^{n_z+2n_\gamma} + \phi_s'\beta_{sv}'(\rho')^2 N^{n_s+1+2n_\rho} + \phi_v'c'\gamma'N^{n_v+n_\gamma} \leq \phi_v'\frac{\mu_G}{16}\rho'N^{n_v+n_\rho}$$

$$5\phi_z'L_z'(\rho')^2 N^{n_z+2n_\rho} + 5\phi_v'L_v'(\rho')^2 N^{n_v+2n_\rho} + L^\Phi L_x'(\gamma')^2 N^{2n_\gamma} + \phi_s'c'\gamma'N^{n_s+n_\gamma} \leq \phi_s'\frac{\Gamma'}{2}N^{n_s-1} \ .$$

In order to ensure that the exponents on $N$ are lower in the left-hand-side than those on the right-hand-side, we take $n_z = n_v = 0$, $n_\rho = -\frac{2}{3}$, $n_\gamma = -1$ and $n_s = -\frac{1}{3}$. The Equations become

$$c' \leq \mu_\Phi$$

$$2\phi_z'\overline{\beta}_{zx}\frac{\gamma'}{\rho'}N^{-\frac{1}{3}} + 2\phi_v'\overline{\beta}_{vx}\frac{\gamma'}{\rho'}N^{-\frac{1}{3}} + \phi_s'P'\gamma'N^{-\frac{1}{3}} + c' \leq \frac{1}{4}$$

$$L_x^2\gamma'N^{-1} + 8\phi_v'\beta_{vz}\rho'N^{-\frac{2}{3}} + \phi_s'\beta_{sz}'(\rho')^2 N^{-\frac{2}{3}} + \phi_z'c'\gamma'N^{-1} \leq \phi_z'\frac{\mu_G}{8}\rho'N^{-\frac{2}{3}}$$

$$L_x^2\gamma'N^{-1} + 8\phi_z'L_x''(\gamma')^2 N^{-2} + \phi_s'\beta_{sv}'(\rho')^2 N^{-\frac{2}{3}} + \phi_v'c'\gamma'N^{-1} \leq \phi_v'\frac{\mu_G}{16}\rho'N^{-\frac{2}{3}}$$

$$5\phi_z'L_z'(\rho')^2 N^{-\frac{4}{3}} + 5\phi_v'L_v'(\rho')^2 N^{-2} + L^\Phi L_x'(\gamma')^2 N^{-2} + \phi_s'c'\gamma'N^{-\frac{4}{3}} \leq \phi_s'\frac{\Gamma'}{2}N^{-\frac{4}{3}} \ .$$

Now we have to find $\rho'$, $\gamma'$, $\phi'_z$, $\phi'_v$ and $\phi'_s$ that verifies the following conditions (which are a bit stronger than thoose in the previous Equations):

$$c' \leq \mu_\Phi$$

$$2\phi'_z \overline{\beta}_{zx} \frac{\gamma'}{\rho'} + 2\phi'_v \overline{\beta}_{vx} \frac{\gamma'}{\rho'} + \phi'_s P' \gamma' + c' \leq \frac{1}{4}$$

$$L_x^2 \gamma' + 8\phi'_v \beta_{vz} \rho' + \phi'_s \beta'_{sz} (\rho')^2 + \phi'_z c' \gamma' \leq \phi'_z \frac{\mu_G}{8} \rho'$$

$$L_x^2 \gamma' + 8\phi'_z L''_x (\gamma')^2 + \phi'_s \beta'_{sv} (\rho')^2 + \phi'_v c' \gamma' \leq \phi'_v \frac{\mu_G}{16} \rho'$$

$$5\phi'_z L'_z (\rho')^2 + 5\phi'_v L'_v (\rho')^2 + L^\Phi L'_x (\gamma')^2 + \phi'_s c' \gamma' \leq \phi'_s \frac{\Gamma'}{2} \ .$$

As previously, we take $\phi'_s = 1$ and we denote $\phi'_z = \phi''_z \frac{\rho'}{\gamma'}$ with $\phi''_z = \frac{1}{32\overline{\beta}_{zx}}$ and $\phi'_z = \phi''_z \frac{\rho'}{\gamma'}$ with $\phi''_v = \min\left(\frac{1}{32\overline{\beta}_{vx}}, \phi''_z \frac{\mu_G}{128\beta_{vz}}\right)$, the equations become

$$c' \leq \mu_\Phi$$

$$P' \gamma' + c' \leq \frac{1}{8}$$

$$L_x^2 (\gamma')^2 + \beta'_{sz} (\rho')^2 \gamma' + \phi''_z c' \rho' \gamma' \leq \phi''_z \frac{\mu_G}{16} (\rho')^2$$

$$L_x^2 (\gamma')^2 + 8\phi''_z L''_x \rho' (\gamma')^2 + \beta'_{sv} (\rho')^2 \gamma' + \phi''_v c' \rho' \gamma' \leq \phi''_v \frac{\mu_G}{16} (\rho')^2$$

$$5\phi''_z L'_z (\rho')^3 + 5\phi''_v L'_v (\rho')^3 + L^h L'_x (\gamma')^3 + c' (\gamma')^2 \leq \frac{\Gamma'}{2} \gamma' \ .$$

Since $c' \leq \frac{1}{16}$ and $\gamma' \leq \frac{1}{16P'}$, the second equation is verified. With $\gamma' \leq \min\left(\sqrt{\frac{\phi''_z \mu_G}{48L_x^2}} \rho', \frac{\phi''_z \mu_G}{48\beta_{sv}}\right)$ and $c' \leq \frac{\mu_G \rho'}{48\gamma'}$ the third is verified. The conditions $\gamma' \leq \min\left(\sqrt{\frac{\phi''_v \mu_G}{64L_x^2}} \rho', \sqrt{\frac{\phi''_v \mu_G}{512\phi''_z L''_x \rho'}}, \frac{\phi''_v \mu_G}{64\beta'_{sv}}\right)$ and $c' \leq \frac{\mu_G \rho'}{64\gamma'}$ ensure that the forth is verified. With $\gamma' \leq \sqrt{\frac{\Gamma'}{8L^\Phi L'_x}}$ and $c' \leq \frac{\Gamma'}{8\gamma'}$, the fifth is simplified in

$$5\phi''_z L'_z (\rho')^3 + 5\phi''_v L'_v (\rho')^3 \leq \frac{\Gamma'}{4} \gamma' \ .$$

Let us denote $\gamma' = \xi\rho'$. To verify this equation and the previous bounds on $\gamma'$ and $c'$, we need

$$\gamma' \leq \underbrace{\min\left(\sqrt{\frac{\phi_z''\mu_G}{48L_x^2}}, \sqrt{\frac{\phi_v''\mu_G}{64L_x^2}}, \sqrt{\frac{L_z'}{2L_x'\beta_{zx}}}, \sqrt{\frac{L_v'}{2L_x'\beta_{zx}}}\right)}_{K_1}\rho' \ ,$$

$$\gamma' \leq \underbrace{\min\left(\sqrt{\frac{\mu_G}{64\beta_{zx}L_x''}}, \sqrt{\frac{\mu_G}{128\beta_{vx}L_x''}}, \sqrt{\frac{\beta_{vz}}{4L_x''\beta_{vx}}}\right)}_{K_2}\sqrt{\rho'} \ ,$$

$$\gamma' \leq \underbrace{\sqrt{\frac{\phi_v''\mu_G}{512\phi_z''L_x''}}}_{K_3}\frac{1}{\sqrt{\rho'}} \ ,$$

$$\gamma' \leq \underbrace{\min\left(\frac{1}{4L^\Phi}, \frac{L_x^2}{2L^\Phi L_x''}, \frac{\phi_z''\mu_G}{48\beta_{sv}}, \frac{\phi_v''\mu_G}{64\beta_{sv}'}, \frac{1}{16P'}, \sqrt{\frac{\Gamma'}{8L^hL_x'}}\right)}_{K_4}$$

$$\gamma' \geq \underbrace{\frac{20(\phi_z''L_z' + \phi_v''L_v')}{20}}_{K_5}(\rho')^3 \ ,$$

$$c' \leq \underbrace{\min\left(\mu_\Phi, \frac{1}{16}, \frac{1}{16P'}\right)}_{K_6} \ ,$$

$$c' \leq \underbrace{\frac{\mu_G}{64}}_{K_7}\frac{1}{\xi} \ ,$$

$$c' \leq \underbrace{\frac{\Gamma'}{8}}_{K_8}\frac{1}{\gamma'} \ .$$

So, $\xi$, $\rho'$ and $c'$ must verify

$$\xi \leq \underbrace{\min\left(K_1, \frac{K_7}{c'}\right)}_{K_1'} \ ,$$

$$\xi \leq K_2(\rho')^{-\frac{1}{2}} \ ,$$

$$\xi \leq K_3(\rho')^{-\frac{3}{2}} \ ,$$

$$\xi \leq \underbrace{\min\left(K_4, \frac{K_8}{c'}\right)}_{K_4'}(\rho')^{-1}$$

$$\xi \geq K_5(\rho')^2 \ ,$$

$$c' \leq \underbrace{\min\left(\mu_\Phi, \frac{1}{16}, \frac{1}{16P'}\right)}_{K_6} \ ,$$

which is possible if

$$\rho' \leq \min\left(\sqrt{\frac{K_1'}{K_5}}, \left(\frac{K_2}{K_5}\right)^{\frac{2}{5}}, \left(\frac{K_3}{K_5}\right)^{\frac{2}{7}}, \left(\frac{K_4'}{K_5}\right)^{\frac{1}{3}}\right) \ .$$

So let us take $c' = \min\left(\mu_\Phi, \frac{1}{16}, \frac{1}{16P'}\right) = \min\left(\mu_\Phi, \frac{1}{16P'}\right)$,

$$\rho' = \min\left(\sqrt{\frac{K_1'}{K_5}}, \left(\frac{K_2}{K_5}\right)^{\frac{2}{5}}, \left(\frac{K_3}{K_5}\right)^{\frac{2}{7}}, \left(\frac{K_4'}{K_5}\right)^{\frac{1}{3}}, \frac{\mu_G}{64L_z^2}, \frac{\overline{\beta}_{zx}}{2\beta_{zx}}, \frac{\mu_G}{128(L_v^2 + L_v'')}, \frac{\beta_{vz}}{8(L_v^2 + L_v'')}, \frac{\overline{\beta}_{vx}}{2\beta_{vx}}\right)$$

and

$$\xi = \min(K_1, K_2(\rho')^{-\frac{1}{2}}, K_3(\rho')^{-\frac{3}{2}}, K_4(\rho')^{-1}) \ .$$

We have

$$\mathcal{L}^{t+1} \le (1 - c')\mathcal{L}^t$$

therefore, unrolling yields

$$\Phi^t - \Phi^* \le \mathcal{L}^t \le (1 - c'\gamma')^t \mathcal{L}^0,$$

∎

