# OpenReview forum: "Efficient Fully Single-Loop Variance Reduced Methods for Stochastic Bilevel Optimization"
_ICLR.cc/2024/Conference — Submitted to ICLR 2024_

### Official Review · Reviewer_oXEw · 2023-10-27

**Soundness:** 3 good
**Presentation:** 3 good
**Contribution:** 3 good
**Rating:** 6
**Confidence:** 4

**Summary:**

This paper considers the stochastic bilevel optimization problem and proposes a new fully single-loop method using the LSVRG to approximate the gradient. Also, theoretical analysis and experiments are presented to show the superiority of the proposed method.

**Strengths:**

1. This paper is well-written and easy to follow.
2. The proposed is novel and the theoretical analysis and experiments are presented to show the superiority of the proposed method.

**Weaknesses:**

1. More experiments are expected, such as meta-learning, and poison attack. I think hyperparameters selection and data hyper-cleaning are somewhat similar, experiments on other applications are expected.
2. Some O(1) sample complexity methods should be compared, e.g. SUSTAIN[1]
3. The proposed method seems can not effectively solve the large-scale problem. Can the author give some results on the large-scale datasets?
4. The convergence analysis is based on the PL condition. An analysis on a more general case is expected.

[1] Khanduri P, Zeng S, Hong M, et al. A near-optimal algorithm for stochastic bilevel optimization via double-momentum[J]. Advances in neural information processing systems, 2021, 34: 30271-30283.

**Questions:**

1. Can the author explain the relation between the gradient estimation of the proposed method and other hypergradient methods?
2. Why the complexity of SVRB is different from VRBO in Table 1? I think they have the same complexity since they all use STORM.
See other questions in weakness.

On page 4, below Equation (4), the sentence is not correct.

---

> ### Author Response · Authors · 2023-11-21
> **Response to Reviewer oXEw (Part 1)**
>
> Thank you for your valuable comments. We hope our response below can adequately address your concerns.
>
> ---
> **Q1**: More experiments are expected, such as meta-learning, and poison attack. I think hyperparameters selection and data hyper-cleaning are somewhat similar, experiments on other applications are expected. **W3**: The proposed method seems can not effectively solve the large-scale problem. Can the author give some results on the large-scale datasets?
>
> **A1**: We appreciate the reviewer's interest in further experimental validation. Our study primarily focuses on theoretical aspects, as detailed in the general response to all reviewers under the section 'Novelty and Significance of Our Work'. In Sections 4.1 and 4.2, we explore hyperparameter selection and data hyper-cleaning, respectively, which are distinct yet popular tasks. To demonstrate the efficacy of our proposed SBO-LSVRG, we conducted experiments using the Convtype and MNIST datasets, which vary significantly in scale and distribution.The limited time frame of the rebuttal period restricts our ability to conduct additional experiments on new datasets, to compare with more baselines, to verify the correctness of our implementation, and to analyze new results. However, we are keen on pursuing further experimental investigations in the future.
>
> ---
> **W2**: Some O(1) sample complexity methods should be compared, e.g. SUSTAIN[1].
>
> **A2**: We would like to clarify that when SUSTAIN states a $\mathcal{O}(1)$ sample complexity, it refers to the complexity *per iteration*. We compare our proposed SBO-LSVRG with SUSTAIN[1] as follows:
>
> - Our iteration complexity and sampling complexity are superior. SUSTAIN[1] exhibits a $\mathcal{O}(\epsilon^{-3/2})$ iteration complexity and sampling complexity, whereas ours are both $\mathcal{O}(\epsilon^{-1})$, offering a significant improvement.
>
> - Both methods are single-loop. However, besides the shared assumptions, SUSTAIN assumes boundedness in the inner-level function, specifically $\|\nabla_{xy}^2 g(x, y)\|^2 \leq C_{g_{xy}}$ (Assumption 2.v).
>
> - SUSTAIN employs a double-momentum acceleration for updating both the inner variable $y$ and the outer variable $x$ (as per Equations 13 and 14 in SUSTAIN). In contrast, our SBO-LSVRG utilizes standard SGD for updating $y$ and $x$ (see Algorithm 1, Lines 5-7 in SBO-LSVRG). There is potential to further enhance the results of SBO-LSVRG by incorporating acceleration methods such as momentum and heavy-ball acceleration.
>
> [1] Khanduri P, Zeng S, Hong M, et al. "A near-optimal algorithm for stochastic bilevel optimization via double-momentum." Advances in neural information processing systems, 2021, 34: 30271-30283.
>
> ---
> **W4**: The convergence analysis is based on the PL condition. An analysis on a more general case is expected.
>
> **A3**: There seems to be a misunderstanding. In our work, we provide convergence analysis for two scenarios: the nonconvex-strongly-convex setting, where the outer-level function is nonconvex and the inner-level function is strongly convex (as presented in Theorem 1), and the scenario under the PL assumption, detailed in Theorem 2.
>
> We also would like to highlight that the PL condition is weaker than strong convexity because it makes less stringent assumptions about the function's curvature and is applicable to a wider range of functions, including those that are not strongly convex. This broader applicability makes the PL condition a useful tool in analyzing the convergence of optimization algorithms, especially in machine learning contexts where the objective functions may not be strongly convex.
>
> ---
> **Q1**: Can the author explain the relation between the gradient estimation of the proposed method and other hypergradient methods?
>
> **A4**: Constructing an efficient hyper-gradient estimator, utilizing the implicit function theorem, is central to existing bilevel optimization methods. The estimation of the Hessian inverse is particularly crucial. For two-loop methods (referenced in the related work section), the typical approach involves first computing the exact or estimated inner optimal $y^{\star}(x)$, followed by employing techniques such as iterative differentiation (ITD), approximate implicit differentiation (AID), and Neumann series-based approximation to approximate the Hessian inverse. In contrast, single-loop methods update each variable simultaneously, necessitating slightly different strategies for Hessian inverse estimation. For instance, inspired by SABA, our approach involves decomposing the variables $ y$ and $x$ and introducing an intermediate variable $v$, derived from the implicit function theorem. These variables are then updated according to their respective derived directions.

---

> > ### Author Response · Authors · 2023-11-21
> > **Response to Reviewer oXEw (Part 2)**
> >
> > **Q2**: Why the complexity of SVRB is different from VRBO in Table 1? I think they have the same complexity since they all use STORM.
> >
> > **A5**: We would like to address and correct an error identified in our previous paper. In particular, we clarify that VRBO incorporates SARAH[a] as its momentum mechanism, in contrast to SVRB, which employs STORM. To provide a comprehensive understanding, we have included a detailed comparison of these two variance reduction techniques in Appendix A. Additionally, it is important to note that even when employing the same variance reduction method, convergence results can vary significantly. This variation can be attributed to factors such as the design of double-loop/single-loop structures, the use of different types of momentum (such as heavy-ball or Nesterov acceleration), the implementation of a warm start, and variations in proof techniques and logical approaches.
> >
> > [a] Nguyen, Lam M., Jie Liu, Katya Scheinberg, and Martin Takáč. "SARAH: A novel method for machine learning problems using stochastic recursive gradient." In International conference on machine learning, pp. 2613-2621. PMLR, 2017.
> >
> > ---
> > **Q3**: On page 4, below Equation (4), the sentence is not correct.
> >
> > **A6**: Thank you for pointing this out. We fixed this in the revised version.

---

> > > ### Comment · Reviewer_oXEw · 2023-11-22
> > > **respone**
> > >
> > > I would like to thank the authors' detailed responses and explanations which have addressed my previous concerns.

---

### Official Review · Reviewer_UgVc · 2023-10-27

**Soundness:** 3 good
**Presentation:** 2 fair
**Contribution:** 2 fair
**Rating:** 6
**Confidence:** 4

**Summary:**

The paper proposed a single-loop algorithm inspired by L-SVRG solving (finite-sum) stochastic bilevel optimization problem with an iteration complexity of $\mathcal{O}((m+n)^{3/2}/T )$ and a memory cost of $\mathcal{O}(d + p)$. The main contribution is reducing the memory cost from $\mathcal{O}((m+n)(d+p))$ to $\mathcal{O}(d + p)$ but achieving the same iteration complexity compared with the state-of-the-art algorithms.

**Strengths:**

Originality: In this paper, the authors considered using L-SVRG for problems with a nested structure and proposed a method for solve stochastic bilevel optimization problem based on it.

Quality: Compared with SABA, their approach reduced the memory cost significantly.

Clarity: The overall structure and presentation of the paper is clear.

Significance: This research provide lower memory cost without affecting iteration complexity for stochastic bilevel optimization and other related problems.

**Weaknesses:**

1. In Contribution (c), the authors stated "We establish the link between our method and related areas, such as federated learning and minimax optimization, and we provide a theoretical analysis for both of these areas." However, I don't see any theoretical analysis about federated learning. It is only in the future work section.

2. From my perspective, the novelty of the paper is limited. The only improvement is reducing cost memory by using a different variance reduction technique.

3. The plots in the paper are hard to read. For example, in figure 2, what is the x-axis of the plots. It is better to provide some plots in terms of running time.

**Questions:**

1. In Theorem 1, the authors stated "This result leads to the convergence rate $\mathcal{O}(\epsilon^{−1})$, which is optimal in stochastic bilevel optimization. " I think this result is for general stochastic bilevel optimization problem. But in this paper, the authors considered a finite-sum version of it, which is easily than the general version. The convergence rate could be potentially improved. It would be more convincing if the authors point out more related references.

2. In Corollary 1, the authors stated "the rate under nonconvex conditions remains unclear. We initially introduce a rate of $\mathcal{O}(n^{2/3}\epsilon^{−1})$". But the authors did not state what kind of convergence criteria do they consider here? Can you provide more references related to the single-level result here?

3. In Corollary 2, do you assume $F$ is convex? Or it could be possibly non-convex. If it is non-convex, how do you get the rate $\mathcal{O}(n^{2/3}\epsilon^{-1})$?

---

> ### Author Response · Authors · 2023-11-21
> **Response to Reviewer UgVc (Part 1)**
>
> Thank you so much for your valuable feedback! We sincerely hope our response below can address your concerns.
>
> ---
> **W1**: In Contribution (c), the authors stated "We establish the link between our method and related areas, such as federated learning and minimax optimization, and we provide a theoretical analysis for both of these areas." However, I don't see any theoretical analysis about federated learning. It is only in the future work section.
>
> **A1**: Corollary 1 addresses the convergence rate of single-level optimization in the finite-sum form, which aligns precisely with the standard model used in federated learning. We apologize for any confusion and have emphasized the connection between Corollary 1 and federated learning in the revised version of our paper.
>
> ---
> **W2**: From my perspective, the novelty of the paper is limited. The only improvement is reducing cost memory by using a different variance reduction technique.
>
> **A2**: Thank you for raising this question. To address your concern, we have detailed our primary contributions in the section "Novelty and Significance of Our Work," which can be found in our gneral response to the reviewers.
>
> ---
> **W3**: The plots in the paper are hard to read. For example, in figure 2, what is the x-axis of the plots. It is better to provide some plots in terms of running time.
>
> **A3**: The x-axis represents the oracle, which is the sample complexity weighted by memory space complexity. We fixed the missing x-axis of Figure 2 of our revised paper. In Figure 1, we demonstrate that our proposed SBO-LSVRG not only achieves a better convergence rate but also converges more effectively than the baselines when considering either the number of iterations or the oracle. Although we did not enhance the original convergence rate of SABA, it becomes particularly interesting to examine the interplay between sample complexity and memory complexity, as depicted in Figure 2.
>
> ---
> **Q1**: In Theorem 1, the authors stated "This result leads to the convergence rate $\mathcal{O}(\epsilon^{-1})$, which is optimal in stochastic bilevel optimization. " I think this result is for general stochastic bilevel optimization problem. But in this paper, the authors considered a finite-sum version of it, which is easily than the general version. The convergence rate could be potentially improved. It would be more convincing if the authors point out more related references.
>
> **A4**: The convergence rate of $\mathcal{O}(\epsilon^{-1})$ is optimal and cannot be improved for standard bilevel finite-sum optimization problems. For single-level nonconvex finite-sum optimization, the optimal complexity for finding an $\epsilon$-stationary point $\hat{x}$ such that $\mathbb{E}[\|\nabla f(\hat{x})\|^2] \leq \epsilon$ is $\mathcal{O}(n + n^{1/2}\epsilon^{-1})$ [a, b]. Particularly when $\epsilon$ is the dominant factor, this optimal rate simplifies to $\mathcal{O}(n^{1/2}\epsilon^{-1})$. Considering that finite-sum bilevel optimization encompasses a more general case, with the single-level optimization representing a specific instance, we can confidently state that our proposed rate of $\mathcal{O}(\epsilon^{-1})$ represents the optimal rate in terms of $\epsilon$.
>
> [a] Fang, Cong, Chris Junchi Li, Zhouchen Lin, and Tong Zhang. "Spider: Near-optimal non-convex optimization via stochastic path-integrated differential estimator." Advances in neural information processing systems 31 (2018).
>
> [b] Li, Zhize, Hongyan Bao, Xiangliang Zhang, and Peter Richt\'arik. "PAGE: A simple and optimal probabilistic gradient estimator for nonconvex optimization." In International conference on machine learning, pp. 6286-6295. PMLR, 2021.

---

> ### Author Response · Authors · 2023-11-21
> **Response to Reviewer UgVc (Part 2)**
>
> **Q2**: In Corollary 1, the authors stated "the rate under nonconvex conditions remains unclear. We initially introduce a rate of $\mathcal{O}(n^{2/3}\epsilon^{-1})$. But the authors did not state what kind of convergence criteria do they consider here? Can you provide more references related to the single-level result here?
>
> **A5**: We wish to emphasize that in the original L-SVRG paper [a], the provided convergence rate is $\mathcal{O}\left( \max\lbrace\frac{L_{\max}}{\mu}, n \rbrace \log \epsilon^{-1}\right)$ for strongly-convex single-level finite-sum optimization. During this rebuttal period, we found deep inside the general framework of [b], a convergence rate with $\mathcal{O}(n^{2/3}\epsilon^{-1})$ was derived for single-level finite-sum optimization, which matches our rate in Corollary 1. We added this reference in our revised paper. In our paper, as well as in [b], we adhere to the standard convergence criteria used in nonconvex analysis. This involves finding an $\epsilon$-stationary point where $\mathbb{E}[\|\nabla f(x)\|^2] \leq \epsilon$. We note that some references employ a different bound, $\mathbb{E}[\|\nabla f(x)\|^2] \leq \epsilon^2$. To maintain consistency, we have adjusted these results to align with the $\epsilon$-based bound.
>
> Our adaptation from bilevel to single-level is achieved by setting $\rho^t \equiv 0$, thereby $y, x$ are always constant and we are optimizing the single variable $x$ under the nonconvex case.
>
> [a] Dmitry Kovalev, Samuel Horvath, and Peter Richtarik. "Don't jump through hoops and remove those loops: Svrg and Katyusha are better without the outer loop." In Algorithmic Learning Theory, pp. 451–467. PMLR, 2020
>
> [b] Li, Zhize, and Peter Richtarik. "A unified analysis of stochastic gradient methods for nonconvex federated optimization." arXiv preprint arXiv:2006.07013 (2020).
>
> ---
> **Q3**: In Corollary 2, do you assume $F$ is convex? Or it could be possibly non-convex. If it is non-convex, how do you get the rate $\mathcal{O}(n^{2/3}\epsilon^{-1})$
>
> **A6**: We apologize for any confusion caused. Given that $G$ is strongly convex, our objective in minimax optimization also maintains strong convexity. We have explicitly stated this assumption in Corollary 2 of our revised manuscript.

---

> > ### Comment · Reviewer_UgVc · 2023-11-23
> > **Response**
> >
> > Thank the authors for the nice response. My concerns are addressed. I increase my score to 6.

---

### Official Review · Reviewer_L11w · 2023-10-30

**Soundness:** 3 good
**Presentation:** 3 good
**Contribution:** 2 fair
**Rating:** 6
**Confidence:** 4

**Summary:**

This work provides a study for stochastic bilevel optimization and provided a fully single-loop stochastic bilevel algorithm using an idea from Loopless-SVRG. Compared to SABA, another fully single-loop stochastic optimizer via SAG, the proposed SBO-LSVRG method achieves a similar sample complexity but with less cost in memory and space. The algorithm and analysis are also applied to minimax problem as well. Experiments are provided to demonstrate the effectiveness.

**Strengths:**

1.	The motivation is clear and the study on fully single-loop bilevel method is very important given its simple structure and implementation. Applying the idea from loopless-SVRG is a good contribution, and the authors have done a good job in the algorithmic design and literature review.

2.	The proposed algorithm achieves the same sample complexity as SABA, the best fully single-loop stochastic bilevel method, but with much less memory cost. In terms of the performance, it seems the method can be more efficient than SABA.

**Weaknesses:**

1.	Applying the idea of L-SVRG in bilevel optimization sounds a little bit incremental. However, I feel that some challenges such as the probabilistic selection step and the proof of variance reduction may introduce some new analysis and designs. I strongly suggest that the authors can explicitly point them out instead of just saying “This approach is far from trivial”.

2.	The comparison miss an important baseline as show here (https://arxiv.org/pdf/2302.08766.pdf): it proposes a single-loop SARAH-based bilevel optimizer named SRBA, which achieves a near-optimal $(n+m)^{1/2}\epsilon^{-1}$ sample complexity. It would be good to have a comparison here.

**Questions:**

Overall, I think this work has provided some interesting approach based on the idea from L-SVRG. I like the method and give 6. However, I cannot give higher score given several questions regarding the novelty clarification and the missing baseline. The questions and suggestions can be found in the weakness part.

---

> ### Author Response · Authors · 2023-11-21
> **Response to Reviewer L11w**
>
> We thank you for the valuable review and for sharing relevant references. We hope that our following discussion will address your raised weaknesses and comments.
>
> ---
> **W1**: Applying the idea of L-SVRG in bilevel optimization sounds a little bit incremental. However, I feel that some challenges such as the probabilistic selection step and the proof of variance reduction may introduce some new analysis and designs. I strongly suggest that the authors can explicitly point them out instead of just saying “This approach is far from trivial”.
>
> **A1**: Thank you for raising this question. We have highlighted our key contributions in the section titled "Novelty and Significance of Our Work," which is included in our general response to all reviewers.
>
> ---
> **W2**: The comparison miss an important baseline as show here ([a]): it proposes a single-loop SARAH-based bilevel optimizer named SRBA, which achieves a near-optimal $(m+n)^{1/2}\epsilon^{-1}$ sample complexity. It would be good to have a comparison here.
>
> [a] Dagréou, Mathieu, Thomas Moreau, Samuel Vaiter, and Pierre Ablin. "A Lower Bound and a Near-Optimal Algorithm for Bilevel Empirical Risk Minimization." arXiv e-prints (2023): arXiv-2302.
>
> **A2**: Thank you for providing the reference. We have added it as a reference in the revised paper. Below we would like to compare SRBA and point out the differences:
>
> - **Distinct Motivations**: While SBO-LSVRG and SRBA both build upon the single-level optimization framework proposed by Dagreou et al., 2022, their motivations differ significantly. SARAH is a recursive method with analysis centered on the benefits of such recursive designs, hoping to do better convergence results. In contrast, our approach was developed in response to the high memory space consumption observed in SAGA, aiming to propose a method that reduces this while remaining unaffected by data size scaling.
>
> - **Simplicity and Efficiency**: Our method outperforms SRBA in simplicity and computational efficiency. SARAH-type methods, like SRBA, require multiple inner recursive steps in each iteration and computation of the full gradient over all local functions, which is inefficient. Our L-SVRG approach, however, eliminates the need for a global loop to compute the full gradient and the multiple inner recursive steps, leading to a sample complexity per iteration of $\mathcal{O}(1)$.
>
> - **Reduced Sample Complexity when $m+n$ Is Large**: SRBA's recursive design results in high sample complexity per iteration. Setting the number of inner iterations $p-1 = m+n -1$ yields a sample complexity per iteration of $\mathcal{O}(m+n)$, leading to a total sample complexity at a stationary point of $\mathcal{O}((m+n)^{1/2}\epsilon^{-1}\lor (m+n))$ as detailed in Corollary 3.6 in SRBA. In practical scenarios with significantly large $m$ or $n$, the sample complexity becomes $\mathcal{O}(m+n)$. In contrast, our method achieves a sample complexity of $\mathcal{O}((m+n)^{2/3}\epsilon^{-1})$. Especially when $m$ or $n$ is large, our method proves more efficient by a factor of $(m+n)^{1/6}$. Furthermore, SRBA's analysis right below Corollary 3.6 reveals that for large $m+n$, the sample complexity approaches $\mathcal{O}(\epsilon^{-2})$, which is suboptimal with respect to $\epsilon$. In conclusion, as $m+n$ becomes large (indicating an increase in data volume), the memory space required in SABA for storing additional status vectors becomes substantial and increases linearly. Meanwhile, SRBA's sample complexity deviates from the optimal $\mathcal{O}(\epsilon^{-1})$. Our method, in contrast, is more practical for large-scale data scenarios and effectively addresses these issues.

---

### Official Review · Reviewer_vkov · 2023-11-01

**Soundness:** 3 good
**Presentation:** 3 good
**Contribution:** 2 fair
**Rating:** 5
**Confidence:** 4

**Summary:**

This paper investigates the problem of bilevel optimization, and introduce  a new method named SBO-LSVRG. This method achieves the SOTA iteration complexity with a lower memory cost. The experiments confirms the effectiveness of the proposed method.

**Strengths:**

1. The paper is well-organized, and easy for readers to follow.
2. This paper can obtain SOTA complexity with lower memery cost. The rigorous proof is provided.

**Weaknesses:**

1. The novelty is limited. This paper mainly follows Dagreou et al. (2022). Therefore, the theoretical contribution is limited. It is better if the authors could highlight the challenges in the analysis.
2. Some assumptions made in the paper seems quite strong, would it hold in practical scenarios?

**Questions:**

1. Can the proposed method handle non-convex or non-strongly convex lower-level problems, which are common in many real-world applications?
2. This paper investigate the iteration complexity of the proposed method, how about the sample compexity?
3. Is the rate obtianed in this paper optimal in terms of $\epsilon$, $m$ and $n$?

---

> ### Author Response · Authors · 2023-11-21
> **Response to Reviewer vkov**
>
> Thank you for taking the time to review our paper and for your thoughtful comments. We hope the following responses can adequately address your concerns.
>
> ---
> **W1**: The novelty is limited. This paper mainly follows Dagreou et al. (2022). Therefore, the theoretical contribution is limited. It is better if the authors could highlight the challenges in the analysis.
>
> **A1**: Thank you for your question. We have emphasized our key contributions under the section titled "Novelty and Significance of Our Work" in our general response to all reviewers.
>
> ---
> **W2**: Some assumptions made in the paper seems quite strong, would it hold in practical scenarios?
>
> **A2**: Our paper introduces three key assumptions in bilevel optimization, as detailed in Section 2.1. Firstly, we assume that the inner-level function $G$ is strongly convex, while the outer-level function $F$ is allowed to be non-convex. This is a common and practical assumption in bilevel optimization, especially when the inner problem is simple. Secondly, we assume smoothness for both inner- and outer-level functions, a standard and realistic expectation for various functions, including neural networks. Lastly, our approach involves practical variance bounds, ensuring that gradients remain bounded in models ranging from logistic regression to complex neural networks.
>
> ---
> **Q1**: Can the proposed method handle non-convex or non-strongly convex lower-level problems, which are common in many real-world applications?
>
> **A3**: The nonconvex-strongly-convex bilevel setting is standard. However, our work can be safely extended to handle the general convex setting using the regularization idea. The brief idea is to suppose $G(y)$ is a convex function; then $\tilde{G}(y) := G(y) + \frac{\mu}{2}\|y\|_2^2$ is a $\mu$-strongly convex function, satisfying our theoretical framework. A smaller $\mu$ implies that $\tilde{G}(y)$ is closer to the initial problem, but also decreases the strong convexity parameter. We expect by suitably balancing this tradeoff on $\mu$ one can establish  meaningful convergence rates.
>
> The convexity of the inner problem is crucial in our proof, as it allows us to find its global optimum. Addressing a non-convex inner problem is significantly more challenging. Even if the outer-level problem is considered constant, finding the *global optimum* of the inner nonconvex problem is known to be difficult. Unless we are willing to relax to a stationary point (which would change the definition of the bilevel formulation), there is generally no efficient solution method.
>
> ---
> **Q2**: This paper investigates the iteration complexity of the proposed method, how about the sample complexity?
>
> **A4**: The sample complexity of SBO-LSVRG is $\mathcal{O}((m+n)^{2/3}\epsilon^{-1})$. In Section 3.2, we specify that sample $i \in [n]$ is chosen with probability $q_y$ and sample $j \in [m]$ with probability $q_x$. Consider $a, b, c > 0$ as constants, independent of iteration $T$. Therefore, the sample complexity, or oracle calls per iteration, is $\mathcal{O}(aq_y(n-1) + bq_x(m-1) + c)$. Adopting uniform sampling with $q_y= 1/n$ and $q_x=1/m$, the sample complexity per iteration simplifies to $\mathcal{O}(1)$. Given that the total number of iterations is $\mathcal{O}((m+n)^{2/3}\epsilon^{-1})$, the overall sample complexity also aligns with $\mathcal{O}((m+n)^{2/3}\epsilon^{-1})$. This update has been reflected in Section 3.2 of our revised paper.
>
> ---
> **Q3**: Is the rate obtianed in this paper optimal in terms of $\epsilon$, $m$ and $n$?
>
> **A5**: Currently, the lower bound for solving bilevel optimization using general methods remains unclear. Nevertheless, in the context of single-level nonconvex finite-sum optimization, the optimal complexity for finding an $\epsilon$-stationary point $\hat{x}$ such that $\mathbb{E}[\|\nabla f(\hat{x})\|^2] \leq \epsilon$ is $\mathcal{O}(n + n^{1/2}\epsilon^{-1})$ [a, b]. Particularly when $\epsilon$ is the dominant factor, this optimal rate simplifies to $\mathcal{O}(n^{1/2}\epsilon^{-1})$. Given that finite-sum bilevel optimization is a more complex scenario, with the single-level optimization representing a specific instance, we can confidently state that our proposed rate of $\mathcal{O}((m+n)^{2/3}\epsilon^{-1})$ represents the optimal rate in terms of $\epsilon$. However, determining the optimal rate with respect to $m$ and $n$ is still an open question.
>
> [a] Fang, Cong, Chris Junchi Li, Zhouchen Lin, and Tong Zhang. "Spider: Near-optimal non-convex optimization via stochastic path-integrated differential estimator." Advances in neural information processing systems 31 (2018).
>
> [b] Li, Zhize, Hongyan Bao, Xiangliang Zhang, and Peter Richt\'arik. "PAGE: A simple and optimal probabilistic gradient estimator for nonconvex optimization." In International conference on machine learning, pp. 6286-6295. PMLR, 2021.

---

### Author Response · Authors · 2023-11-21
**Novelty and Significance of Our Work**

We would like to highlight our novel contributions as follows:

- Our paper is motivated by observing the superior convergence rate of SABA in Dagreou et al. (2022), despite its high memory cost. We investigated the underlying reasons and discovered that an LSVRG-type design could potentially reduce memory costs while maintaining or improving the convergence rate. Though this concept may seem straightforward when knowing the fundamental reasons behind, we have validated its effectiveness through rigorous proof and extensive experimentation.

- The original memory requirement for the baseline SABA model is $\mathcal{O}((m+n)(d+p))$, which scales linearly with $m$ and $n$, proving impractical for scenarios with large datasets. We have successfully reduced the memory footprint to $\mathcal{O}(d+p)$, ensuring that our model's memory consumption remains constant even with increasing data volumes. This reduction facilitates the application of bilevel optimization in solving more complex problems.

- The proof of SBO-LSVRG's effectiveness is nontrivial. Bilevel optimization, being inherently more complex than single-level optimization, presents significant challenges, where even minor modifications can impede proof completion. Existing methods primarily focus on providing a simpler yet efficient algorithm or incorporating new features like accelerated gradient descent and momentum, which are conceptually simple yet highly effective in bilevel optimization (see Table 1 in Dagreou et al. 2022 as a reference). Our approach, compared to SABA, is simpler and more efficient in algorithm design, and we offer a more accessible proof without the need for matrix-sized local variable storage, which is highly valued in theoretical analysis.

- The seminal work of SVRG, introduced by [1], is fundamental in the field of large-scale optimization. Our approach is based on LSVRG ([2]), a loopless variant of SVRG that eliminates the necessity for full-gradient computation in each global loop. In our research, specifically highlighted in Corollaries 1 and 2, we demonstrate that our proposed SBO-LSVRG is effectively applicable to both single-level finite-sum and minimax optimization, achieving an optimal rate with respect to $\epsilon$. This is particularly significant in areas such as federated learning and generative adversarial training. Further, in our rebuttal, we delved into the details of [3] and discovered that it provides the convergence rate for single-level non-convex LSVRG as $\mathcal{O}(n^{2/3}\epsilon^{-1})$, aligning with what we have presented in our revised paper. Moreover, Corollary 2 presents the first convergence rate of LSVRG-type methods in minimax optimization.

We hope these points clearly articulate the innovative aspects of our work and its significance.

[1] Johnson, Rie, and Tong Zhang. "Accelerating stochastic gradient descent using predictive variance reduction." Advances in neural information processing systems 26 (2013).

[2] Dmitry Kovalev, Samuel Horvath, and Peter Richtarik. "Don't jump through hoops and remove those loops: Svrg and Katyusha are better without the outer loop." In Algorithmic Learning Theory, pp. 451–467. PMLR, 2020

[3] Li, Zhize, and Peter Richtarik. "A unified analysis of stochastic gradient methods for nonconvex federated optimization." arXiv preprint arXiv:2006.07013 (2020).

---

### Meta-Review · Area_Chair_tG6Z · 2023-12-06

**Metareview:**

This paper studies stochastic bilevel optimization. It is observed that in a state-of-the-art work due to Dagreou et al 2022, the memory cost is proportional to the sample size. Building upon the memory-efficient approach of Kovalev et al 2020, the paper introduces a new optimization algorithm whose memory cost is independent of the sample size.

**Strengths**
- The paper presents a new memory-efficient algorithm for stochastic bilevel optimization.
- The paper is well motivated and is easy to follow.

**Weaknesses**
- The primary concern on the paper is the technical novelty and significance. Though the memory efficiency appears important and achieving both good convergence rate and memory efficiency seems new, the major tools are largely borrowed from existing works. In particular, the starting point of the paper is Dagreou et al 2022, which provided a general optimization framework to obtain the convergence rate that this paper follows. On the other hand, it is broadly known that the loopless SVRG algorithm of Kovalev et al 2020 can always be an alternative optimizer that is memory efficient and converges fast. It is thus not surprising that one can combine both algorithms to obtain the main results. In this sense, the results are not significant.
- The scalability of the algorithm to large-scale problems is unclear. This is where the paper is motivated but authors were unable to address.
- The writing of the paper can be improved.

**Suggestions to authors**
- Authors are suggested to clarify the main technical challenges to combine Dagreou et al 2022 and Kovalev et al 2020, as well as to highlight the new techniques that are developed to obtain the main results.
- Authors are suggested to test the algorithm on large-scale data sets.
- Authors are suggested to proofread the manuscript to fix typos and grammatical errors.

**Justification For Why Not Higher Score:**

Both the theory and experiments are insignificant.

**Justification For Why Not Lower Score:**

N/A

---

### Decision · Program_Chairs · 2024-01-16

Reject